# Lower Bounds and Optimal Algorithms for Smooth and Strongly Convex Decentralized Optimization Over Time-Varying Networks

**Dmitry Kovalev**
KAUST*
dakovalev1@gmail.com

**Elnur Gasanov**
KAUST
elnur.gasanov@kaust.edu.sa

**Alexander Gasnikov**
MIPT[†] and ISP RAS[‡]
gasnikov@yandex.ru

**Peter Richtárik**
KAUST
richtarik@gmail.com

## Abstract

We consider the task of minimizing the sum of smooth and strongly convex functions stored in a decentralized manner across the nodes of a communication network whose links are allowed to change in time. We solve two fundamental problems for this task. First, we establish *the first lower bounds* on the number of decentralized communication rounds and the number of local computations required to find an $\epsilon$-accurate solution. Second, we design two *optimal algorithms* that attain these lower bounds: (i) a variant of the recently proposed algorithm ADOM (Kovalev et al., 2021) enhanced via a multi-consensus subroutine, which is optimal in the case when access to the dual gradients is assumed, and (ii) a novel algorithm, called ADOM+, which is optimal in the case when access to the primal gradients is assumed. We corroborate the theoretical efficiency of these algorithms by performing an experimental comparison with existing state-of-the-art methods.

## 1 Introduction

In this work we are solving the decentralized optimization problem

$$\min_{x\in\mathbb{R}^d} \sum_{i=1}^{n} f_i(x), \tag{1}$$

where each function $f_i\colon \mathbb{R}^d \to \mathbb{R}^d$ is stored on a compute node $i \in \{1,\ldots,n\}$. We assume that the nodes are connected through a communication network. Each node can perform local computations based on its local state and data, and can directly communicate with its neighbors only. Further, we assume the functions $f_i$ to be smooth and strongly convex.

Such decentralized optimization problems arise in many applications, including estimation by sensor networks (Rabbat and Nowak, 2004), network resource allocation (Beck et al., 2014), cooperative control (Giselsson et al., 2013), distributed spectrum sensing (Bazerque and Giannakis, 2009) and power system control (Gan et al., 2012). Moreover, problems of this form draw attention of the machine learning community (Scaman et al., 2017), since they cover training of supervised machine

---

*King Abdullah University of Science and Technology, Thuwal, Saudi Arabia.

[†]Moscow Institute of Physics and Technology, Moscow, Russia

[‡]Institute for System Programming of the Russian Academy of Sciences, Research Center for Trusted Artificial Intelligence, Moscow, Russia

35th Conference on Neural Information Processing Systems (NeurIPS 2021).

learning models through empirical risk minimization from the data stored across the nodes of a network as a special case. Finally, while the current federated learning (Konečný et al., 2016; McMahan et al., 2017) systems rely on a star network topology, with a trusted server performing aggregation and coordination placed at the center of the network, advances in decentralized optimization could be useful in new-generation federated learning formulations that would rely on fully decentralized computation (Li et al., 2020).

## 1.1 Time-varying Networks

In this work, we focus on the practically highly relevant and theoretically challenging situation when the *links in the communication network are allowed to change over time*. Such *time-varying networks* (Zadeh, 1961; Kolar et al., 2010) are ubiquitous in many complex systems and practical applications. In sensor networks, for example, changes in the link structure occur when the sensors are in motion, and due to other disturbances in the wireless signal connecting pairs of nodes. We envisage that a similar regime will be supported in future-generation federated learning systems (Konečný et al., 2016; McMahan et al., 2017), where the communication pattern among pairs of mobile devices or mobile devices and edge servers will be dictated by their physical proximity, which naturally changes over time.

## 1.2 Contributions

In this work we present the following key contributions:

1. **Lower bounds.** We establish the *first lower bounds* on decentralized communication and local computation complexities for solving problem (1) over *time-varying networks*. Our results are summarized in Table 1, and detailed in Section 3 (see Theorems 2 and 3 therein).

2. **Optimal algorithms.** Further, we prove that *these bounds are tight* by providing *two new optimal algorithms*[4] which match these lower bounds:

   (i) a variant of the recently proposed algorithm ADOM (Kovalev et al., 2021) enhanced via a multi-consensus subroutine, and

   (ii) a novel algorithm, called ADOM+ (Algorithm 1), also featuring multi-consensus.

   The former method is optimal in the case when access to the dual gradients is assumed, and the latter one is optimal in the case when access to the primal gradients is assumed. See Sections 4 and 5 for details. To the best of our knowledge, ADOM with multi-consensus is the first dual based optimal decentralized algorithm for time-varying networks

3. **Experiments.** Through illustrative numerical experiments (see Section 6, and the extra experiments contained in the appendix) we demonstrate that our methods are *implementable*, and that they *perform competitively* when compared to existing baseline methods APM-C (Rogozin et al., 2020; Li et al., 2018) and Acc-GT (Li and Lin, 2021).

**Related Work.** When the communication network is fixed in time, decentralized distributed optimization in the strongly convex and smooth regime is relatively well studied. In particular, Scaman et al. (2017) established lower decentralized communication and local computation complexities for solving this problem, and proposed an optimal algorithm called MSDA in the case when an access to the dual oracle (gradient of the Fenchel transform of the objective function) is assumed. Under a primal oracle (gradient of the objective function), current state of the art includes the near-optimal algorithms APM-C (Li et al., 2018; Dvinskikh and Gasnikov, 2019) and Mudag (Ye et al., 2020), and a recently proposed optimal algorithm OPAPC (Kovalev et al., 2020).

The situation is worse in the time-varying case. To the best of our knowledge, no lower decentralized communication complexity bound exists for this problem. There are a few linearly-convergent algorithms, such as those of Nedic et al. (2017) and Push-Pull Gradient Method of Pu et al. (2020), that assume a primal oracle, and the dual oracle based algorithm PANDA due to Maros and Jaldén (2018). These algorithms have complicated theoretical analyses, which results in slow convergence rates. There are also several accelerated algorithms, which were originally developed for the fixed

---

[4]As a byproduct of our lower bounds, we show that the recently proposed method Acc-GT of Li and Lin (2021) is also optimal. This method appeared on arXiv in April 2021, at the time when we already had a first draft of this paper, including all results. Their work does not offer any lower bounds.

Table 1: Current theoretical state-of-the-art methods for solving problem (1) over time-varying networks, and our contributions: lower bounds (first lower bounds for this problem), and two new optimal algorithms, ADOM and ADOM+ with multi-consensus.

| Algorithm | Local computation complexity | Decentralized communication complexity | Gradient oracle |
|---|---|---|---|
| **Known Results** | | | |
| APM-C (Rogozin et al., 2020) | $\mathcal{O}\left(\kappa^{1/2}\log\frac{1}{\epsilon}\right)$ | $\mathcal{O}\left(\chi\kappa^{1/2}\log^2\frac{1}{\epsilon}\right)$ | primal |
| Mudag (Ye et al., 2020) | $\mathcal{O}\left(\kappa^{1/2}\log\frac{1}{\epsilon}\right)$ | $\mathcal{O}\left(\chi\kappa^{1/2}\log(\kappa)\log\frac{1}{\epsilon}\right)$ | primal |
| Acc-GT with multi-consensus (Li and Lin, 2021) | $\mathcal{O}\left(\kappa^{1/2}\log\frac{1}{\epsilon}\right)$ | $\mathcal{O}\left(\chi\kappa^{1/2}\log\frac{1}{\epsilon}\right)$ | primal |
| **Our Results** | | | |
| ADOM with multi-consensus This Paper, Theorem 5 | $\mathcal{O}\left(\kappa^{1/2}\log\frac{1}{\epsilon}\right)$ | $\mathcal{O}\left(\chi\kappa^{1/2}\log\frac{1}{\epsilon}\right)$ | dual |
| ADOM+ with multi-consensus This Paper, Theorem 6 | $\mathcal{O}\left(\kappa^{1/2}\log\frac{1}{\epsilon}\right)$ | $\mathcal{O}\left(\chi\kappa^{1/2}\log\frac{1}{\epsilon}\right)$ | primal |
| **Lower Bounds This Paper, Theorems 2 and 3** | $\mathcal{O}\left(\kappa^{1/2}\log\frac{1}{\epsilon}\right)$ | $\mathcal{O}\left(\chi\kappa^{1/2}\log\frac{1}{\epsilon}\right)$ | both |

network case, and can be extended to the time-varying case. These include Acc-DNGD (Qu and Li, 2019), Mudag (Ye et al., 2020) and a variant of APM-C which was extended to the time-varying case by Rogozin et al. (2020). Finally, there are two algorithms with state-of-the-art decentralized communication complexity: a dual based algorithm ADOM (Kovalev et al., 2021), and a primal based algorithm Acc-GT (Li and Lin, 2021).

## 2 Notation and Assumptions

### 2.1 Smooth and Strongly Convex Regime

Throughout this paper we restrict each function $f_i(x)$ to be $L$-smooth and $\mu$-strongly convex. That is, we require the following inequalities to hold for all $x, y \in \mathbb{R}^d$ and $i \in \{1, \ldots, n\}$:

$$f_i(y) + \langle \nabla f_i(y), x - y \rangle + \frac{\mu}{2}\|x - y\|^2 \le f_i(x) \le f_i(y) + \langle \nabla f_i(y), x - y \rangle + \frac{L}{2}\|x - y\|^2. \quad (2)$$

This naturally leads to the quantity $\kappa = \frac{L}{\mu}$ known as the condition number of function $f_i$. Strong convexity implies that problem (1) has a unique solution.

### 2.2 Primal and Dual Oracle

In our work we consider two types of gradient oracles. By *primal oracle* we denote the situation when the gradients $\nabla f_i$ of the objective functions $f_i$ are available. By *dual oracle* we denote the situation when the gradients $\nabla f_i^*$ of the Fenchel conjugates[5] $f_i^*$ of the objective functions $f_i$ are available.

### 2.3 Decentralized Communication

Let $\mathcal{V} = \{1, \ldots, n\}$ denote the set of the compute nodes. We assume that decentralized communication is split into communication rounds. At each round $q \in \{0, 1, 2, \ldots\}$, nodes are connected through a communication network represented by a graph $\mathcal{G}^q = (\mathcal{V}, \mathcal{E}^q)$, where $\mathcal{E}^q \subset \{(i, j) \in \mathcal{V} \times \mathcal{V} : i \ne j\}$ is the set of links at round $q$. For each node $i \in \mathcal{V}$ we consider a set of its immediate neighbors at round $q$: $\mathcal{N}_i^q = \{j \in \mathcal{V} : (i, j) \in \mathcal{E}^q\}$. At round $q$, each node $i \in \mathcal{V}$ can communicate with nodes from the set $\mathcal{N}_i^q$ only. This type of communication is known in the literature as decentralized communication.

---

[5]Recall that the Fenchel conjugate of $f_i$ is given as $f_i^*(x) = \sup_{y \in \mathbb{R}^d}[\langle x, y \rangle - f_i(y)]$. Note, that $f_i^*$ is $1/\mu$-smooth and $1/L$-strongly convex (Rockafellar, 2015).

## 2.4 Gossip Matrices

Decentralized communication between nodes is typically represented via a matrix-vector multiplication with a gossip matrix. For each decentralized communication round $q \in \{0, 1, 2, \ldots\}$, consider a matrix $\mathbf{W}(q) \in \mathbb{R}^{n \times n}$ satisfying the following assumption.

**Assumption 1.** *For any* $q \in \{0, 1, 2, \ldots\}$ *matrix* $\mathbf{W}(q) \in \mathbb{R}^{n \times n}$ *satisfies the following relations:*

1. $\mathbf{W}(q)_{i,j} = 0$ *if* $i \neq j$ *and* $(i, j) \notin \mathcal{E}^q$,

2. $\ker \mathbf{W}(q) \supset \{(x_1, \ldots, x_n) \in \mathbb{R}^n : x_1 = \ldots = x_n\}$,

3. $\mathrm{range}\mathbf{W}(q) \subset \{(x_1, \ldots, x_n) \in \mathbb{R}^n : \sum_{i=1}^{n} x_i = 0\}$,

4. *There exists* $\chi \geq 1$, *such that*
   $$\|\mathbf{W}x - x\|^2 \leq (1 - \chi^{-1})\|x\|^2 \text{ for all } x \in \{(x_1, \ldots, x_n) \in \mathbb{R}^n : \sum_{i=1}^{n} x_i = 0\}. \quad (3)$$

Throughout this paper we will refer to the matrix $\mathbf{W}(q)$ as the *gossip matrix*. A typical example of a gossip matrix is $\mathbf{W}(q) = \lambda_{\max}^{-1}(\mathbf{L}(q)) \cdot \mathbf{L}(q)$, where $\mathbf{L}(q)$ the Laplacian of an undirected connected graph $\mathcal{G}^q$. Note that in this case $\mathbf{W}(q)$ is symmetric and positive semi-definite, and $\chi$ can be chosen as an upper-bound on the condition number of the matrix $\mathbf{L}(q)$ defined by $\chi = \sup_q \frac{\lambda_{\max}(\mathbf{L}(q))}{\lambda_{\min}^+(\mathbf{L}(q))}$, where $\lambda_{\max}(\mathbf{L}(q))$ and $\lambda_{\min}^+(\mathbf{L}(q))$ denote the largest and the smallest positive eigenvalue of $\mathbf{L}(q)$, respectively. With a slight abuse of language, we will call $\chi$ the *condition number of time-varying network* even when the gossip matrices $\mathbf{W}(q)$ are not necessarily symmetric.

While the notion of a gossip matrix is typically used in the fixed network setting (Scaman et al., 2017), we adapt this notion to the time-varying regime in which the so-called mixing matrices (Nedic et al., 2017) are typically used instead. However, it turns out that there is no major difference between these notions. While we decided to present our main results in the language of gossip matrices, they can alternatively be described in the language of mixing matrices instead. We provide a detailed discussion of this in the appendix.

## 3 Lower Complexity Bounds

In this section we obtain lower bounds on the decentralized communication and local computation complexities for solving problem (1). These lower bounds apply to algorithms which belong to a certain class, which we call *First-order Decentralized Algorithms*. Algorithms from this class need to satisfy the following assumptions:

1. Each compute node can calculate first-order characteristics, such as gradient of the function it stores, or its Fenchel conjugate.

2. Each compute node can communicate values, such as vectors from $\mathbb{R}^d$, with its neighbors. Note that the set of neighbors for each node is not fixed in time.

We repeat here that when the network is fixed in time, lower decentralized communication and local computation complexity bounds were obtained by Scaman et al. (2017).

### 3.1 First-order Decentralized Algorithms

Here we give a formal definition of the class of algorithms for which we provide lower bounds on the complexity of solving problem (1). At each time step $k \in \{0, 1, 2, \ldots\}$, each node $i \in \mathcal{V}$ maintains a finite local memory $\mathcal{H}_i(k) \subset \mathbb{R}^d$. For simplicity, we assume that the local memory is initialized as $\mathcal{H}_i(k) = \{0\}$.

At each time step $k$, an algorithm either performs a decentralized communication round, or a local computation round that can update the memory $\mathcal{H}_i(k)$. The update of the local memory satisfies the following rules:

1. If an algorithm performs a local computation round at time step $k$, then
   $$\mathcal{H}_i(k+1) \subset \mathrm{Span}(\{x, \nabla f_i(x), \nabla f_i^*(x) : x \in \mathcal{H}_i(k)\})$$
   for all $i \in \mathcal{V}$, where $f_i^*$ is the Fenchel conjugate of $f_i$.

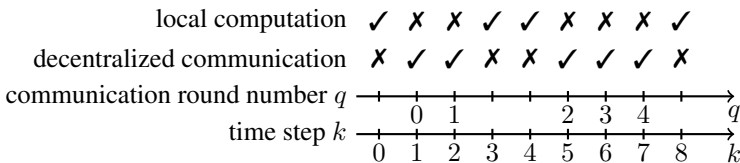

Figure 1: The way we count decentralized communication rounds ($q$) and iterations ($k$).

2. If an algorithm performs a decentralized communication round at time step $k$, then

$$\mathcal{H}_i(k+1) \subset \mathrm{Span}\left(\cup_{j\in\mathcal{N}_i^q\cup\{i\}}\mathcal{H}_j(k)\right)$$

for all $i \in \mathcal{V}$, where $q \in \{0, 1, 2, \ldots\}$ refers to the number of decentralized communication round performed so far, counting from the first one, for which we set $q = 0$. See Figure 1 for a graphical illustration of our time step ($k$) and decentralized communication ($q$) counters.

At time step $k$, each node $i \in \mathcal{V}$ must specify an output value $x_i^o(k) \in \mathcal{H}_i(k)$.

In the case of time-invariant networks, Scaman et al. (2017) presented lower complexity bounds for a class of algorithms called *Black-box Optimization Procedures*. This class is slightly more general than our First-order Decentralized Algorithms. In particular, we use discrete time $k \in \{0, 1, 2, \ldots\}$ rather than continuous time, and do not allow decentralized communications and local computations to be performed in parallel and asynchronously. We have done this for the sake of simplicity and clarity. This should not be seen as a weakness of our work, since our results can be easily extended to Black-box Optimization Procedures. However, we are of the opinion that such an extension would not give any new substantial insights.

## 3.2 Main Theorems

We are now ready to present our main theorems describing the first lower bounds on the number of decentralized communication rounds and local computation rounds that are necessary to find an approximate solution of the problem (1). In order to simplify the proofs, and following the approach used by Nesterov (2003), we consider the limiting case $d \to +\infty$. More precisely, we will work in the infinite dimensional space $\ell_2 = \left\{x = (x_{[l]})_{l=1}^\infty : \sum_{l=1}^\infty (x_{[l]})^2 < +\infty\right\}$ rather than $\mathbb{R}^d$.

**Theorem 1.** *Let $\chi \geq 3$, $L > \mu \geq 0$. There exists a sequence of graphs $\{\mathcal{G}^q\}_{q=0}^\infty$, a corresponding sequence of gossip matrices $\{\mathbf{W}(q)\}_{q=0}^\infty \subset \mathbb{R}^{n\times n}$ satisfying the Assumption 1, and $\mu$-strongly convex and $L$-smooth functions $f_i : \ell_2 \to \mathbb{R}, i \in \mathcal{V}$, such that for any first-order decentralized algorithm and any $k \in \{0, 1, 2\ldots\}$*

$$\|x_i^o(k) - x^*\|^2 \geq C \left(\max\left\{0, 1 - \frac{24\sqrt{6\mu}}{\sqrt{L}}\right\}\right)^{q/\chi}, \tag{4}$$

*where $q$ is the number of decentralized communication rounds performed by the algorithm before time step $k$, $x^*$ is the solution of the problem (1) and $C > 0$ is some constant.*

Our proof is inspired by the proof of Scaman et al. (2017) for time-invariant networks, which in its turn is based on the proof of oracle complexities for strongly convex and smooth optimization by Nesterov (2003). Here we give a short outline of our proof.

We choose $n = |\mathcal{V}| \approx \chi$ nodes and split them into three disjoint sets, $\mathcal{V}_1, \mathcal{V}_2$ and $\mathcal{V}_3$, of equal size $n/3$. Further, we split the function used by Nesterov (2003) onto the nodes belonging to $\mathcal{V}_1$ and $\mathcal{V}_3$. One then needs to show that most dimensions of the output vectors $x_i^o(k)$ will remain zero, while local computations may only increase the number of non-zero dimensions by one. In contrast to (Scaman et al., 2017), we need to have $|\mathcal{V}_2| \approx \chi$ rather than $|\mathcal{V}_2| \approx \sqrt{\chi}$, and still ensure that at least $|\mathcal{V}_2|$ decentralized communication rounds are necessary to share information between node groups $\mathcal{V}_1$ and $\mathcal{V}_3$. We achieve this by choosing a sequence of star graphs with the center node cycling through the nodes from $\mathcal{V}_2$.

**Theorem 2.** *For any $\chi \geq 3$, $L > \mu \geq 0$ there exists a sequence of graphs $\{\mathcal{G}^q\}_{q=0}^\infty$, a corresponding sequence of gossip matrices $\{\mathbf{W}(q)\}_{q=0}^\infty \subset \mathbb{R}^{n\times n}$ satisfying the Assumption 1, and $\mu$-strongly convex*

and L-smooth functions $f_i : \ell_2 \to \mathbb{R}, i \in \mathcal{V}$, such that for any first order decentralized algorithm, the number of decentralized communication rounds to find an $\epsilon$-accurate solution of the problem (1) is lower bounded by

$$\Omega\left(\chi\sqrt{L/\mu}\log\tfrac{1}{\epsilon}\right). \tag{5}$$

We also provide a lower bound on the local computation complexity. The proof is similar to the proof of Theorem 1.

**Theorem 3.** *For any $\chi \geq 3$, $L > \mu \geq 0$ there exists a sequence of graphs $\{\mathcal{G}^q\}_{q=0}^{\infty}$, a corresponding sequence of gossip matrices $\{\mathbf{W}(q)\}_{q=0}^{\infty} \subset \mathbb{R}^{n \times n}$ satisfying the Assumption 1, and $\mu$-strongly convex and L-smooth functions $f_i : \ell_2 \to \mathbb{R}, i \in \mathcal{V}$, such that for any first order decentralized algorithm, the number of local computation rounds to find an $\epsilon$-accurate solution of the problem (1) is lower bounded by*

$$\Omega\left(\sqrt{L/\mu}\log\tfrac{1}{\epsilon}\right). \tag{6}$$

The detailed proofs are available in the appendix.

# 4 Primal Algorithm with Optimal Communication Complexity

In this section we develop a novel algorithm for decentralized optimization over time-varying networks with optimal decentralized communication complexity. This is a primal algorithm, meaning that it uses the primal gradient oracle. The design of this algorithm relies on a sequence of specific reformulations of the problem (1), which we now describe.

## 4.1 Reformulation via Lifting

Consider a function $F \colon (\mathbb{R}^d)^{\mathcal{V}} \to \mathbb{R}$ defined by

$$F(x) = \sum_{i \in \mathcal{V}} f_i(x_i), \tag{7}$$

where $x = (x_1, \ldots, x_n) \in (\mathbb{R}^d)^{\mathcal{V}}$. This function is $L$-smooth and $\mu$-strongly convex since the individual functions $f_i$ are. Consider also the so called consensus space $\mathcal{L} \subset (\mathbb{R}^d)^{\mathcal{V}}$ defined by

$$\mathcal{L} = \{(x_1, \ldots, x_n) \in (\mathbb{R}^d)^{\mathcal{V}} : x_1 = \ldots = x_n\}. \tag{8}$$

Using this notation, we arrive at the equivalent formulation of problem (1)

$$\min_{x \in \mathcal{L}} F(x). \tag{9}$$

Due to strong convexity, this reformulation has a unique solution, which we denote as $x^* \in \mathcal{L}$.

## 4.2 Saddle Point Reformulation

Next, we introduce a reformulation of problem (9) using a parameter $\nu \in (0, \mu)$ and a slack variable $w \in (\mathbb{R}^d)^{\mathcal{V}}$:

$$\min_{\substack{x, w \in (\mathbb{R}^d)^{\mathcal{V}} \\ w = x, w \in \mathcal{L}}} F(x) - \tfrac{\nu}{2}\|x\|^2 + \tfrac{\nu}{2}\|w\|^2.$$

Note that the function $F(x) - \tfrac{\nu}{2}\|x\|^2$ is $(\mu - \nu)$-strongly convex since $\nu < \mu$. The latter problem is a minimization problem with linear constraints. Hence, it has the equivalent saddle-point reformulation

$$\min_{x, w \in (\mathbb{R}^d)^{\mathcal{V}}} \max_{y \in (\mathbb{R}^d)^{\mathcal{V}}} \max_{z \in \mathcal{L}^{\perp}} F(x) - \tfrac{\nu}{2}\|x\|^2 + \tfrac{\nu}{2}\|w\|^2 + \langle y, w - x \rangle + \langle z, w \rangle,$$

where $\mathcal{L}^{\perp} \subset (\mathbb{R}^d)^{\mathcal{V}}$ is an orthogonal complement to the space $\mathcal{L}$, defined by

$$\mathcal{L}^{\perp} = \{(z_1 \ldots, z_n) \in (\mathbb{R}^d)^{\mathcal{V}} : \textstyle\sum_{i=1}^{n} z_i = 0\}. \tag{10}$$

Minimization in $w$ gives the final saddle-point reformulation of the problem (9):

$$\min_{x \in (\mathbb{R}^d)^{\mathcal{V}}} \max_{y \in (\mathbb{R}^d)^{\mathcal{V}}} \max_{z \in \mathcal{L}^{\perp}} F(x) - \tfrac{\nu}{2}\|x\|^2 - \langle y, x \rangle - \tfrac{1}{2\nu}\|y + z\|^2. \tag{11}$$

Further, by $\mathsf{E}$ we denote the Euclidean space $\mathsf{E} = (\mathbb{R}^d)^{\mathcal{V}} \times (\mathbb{R}^d)^{\mathcal{V}} \times \mathcal{L}^{\perp}$. One can show that the saddle-point problem (11) has a unique solution $(x^*, y^*, z^*) \in \mathsf{E}$, which satisfies the following optimality conditions:

$$0 = \nabla F(x^*) - \nu x^* - y^*, \tag{12}$$

$$0 = \nu^{-1}(y^* + z^*) + x^*, \tag{13}$$

$$\mathcal{L} \ni y^* + z^*. \tag{14}$$

### 4.3 Monotone Inclusion Reformulation

Consider two monotone operators $A, B \colon \mathsf{E} \to \mathsf{E}$, defined via

$$A(x, y, z) = \begin{bmatrix} \nabla F(x) - \nu x \\ \nu^{-1}(y + z) \\ \mathbf{P}\nu^{-1}(y + z) \end{bmatrix}, \qquad B(x, y, z) = \begin{bmatrix} -y \\ x \\ 0 \end{bmatrix}, \tag{15}$$

where $\mathbf{P}$ is an orthogonal projection matrix onto the subspace $\mathcal{L}^{\perp}$. Matrix $\mathbf{P}$ is given as

$$\mathbf{P} = (\mathbf{I}_n - \tfrac{1}{n}\mathbf{1}_n\mathbf{1}_n^{\top}) \otimes \mathbf{I}_d, \tag{16}$$

where $\mathbf{I}_p$ denotes $p \times p$ identity matrix, $\mathbf{1}_n = (1, \ldots, 1) \in \mathbb{R}^n$, and $\otimes$ is the Kronecker product. Then, solving problem (11) is equivalent to finding $(x^*, y^*, z^*) \in \mathsf{E}$, such that

$$A(x^*, y^*, z^*) + B(x^*, y^*, z^*) = 0. \tag{17}$$

Indeed, optimality condition (14) is equivalent to $\text{proj}_{\mathcal{L}^{\perp}}(y^* + z^*) = 0$ or $\mathbf{P}\nu^{-1}(y^* + z^*) = 0$. Now, it is clear that (17) is just another way to write the optimality conditions for problem (11).

### 4.4 Primal Algorithm Design and Convergence

A common approach to solving problem (17) is to use the Forward-Backward algorithm. The update rule of this algorithm is $(x^+, y^+, z^+) = J_{\omega B}[(x, y, z) - \omega A(x, y, z)]$, where $\omega > 0$ is a stepsize, the operator $J_{\omega B} \colon \mathsf{E} \to \mathsf{E}$ is called the resolvent, and is defined as the inverse of the operator $(I + \omega B) \colon \mathsf{E} \to \mathsf{E}$, where $I \colon \mathsf{E} \to \mathsf{E}$ is the identity mapping. One can observe that the resolvent $J_{\omega B}$ is easy to compute.

Following (Kovalev et al., 2020), we use an accelerated version of the Forward-Backward algorithm. This can indeed be done, since the operator $A \colon \mathsf{E} \to \mathsf{E}$ is the gradient of the smooth and convex function $(x, y, z) \mapsto F(x) - \frac{\nu}{2}\|x\|^2 + \frac{1}{2\nu}\|y + z\|^2$ on the Euclidean space $\mathsf{E}$. Note that the operator $A + B$ is *not* strongly monotone, while strong monotonicity is usually required to achieve linear convergence. However, we can still obtain a linear convergence rate by carefully utilizing the $L$-smoothness property of the function $F(x)$. Similar issue appeared in the design of algorithms for solving linearly-constrained minimizations problems, including the non-accelerated algorithms (Condat et al., 2019; Salim et al., 2020) and the optimal algorithm (Salim et al., 2021). However, the authors of these works considered a different problem reformulation from (11), and hence their results can not be applied here.

However, one issue still remains: to compute the operator $A$, it is necessary to perform a matrix-vector multiplication with the matrix $\mathbf{P}$, which requires full averaging, i.e, consensus over all nodes of the network. Following the approach of Kovalev et al. (2021), we replace it with the multiplication via the gossip matrix $\mathbf{W}(q) \otimes \mathbf{I}_d$, which requires one decentralized communication round only. That is, we replace the last component $\mathbf{P}\nu^{-1}(y + z)$ of the operator $A$ defined by (15) with $(\mathbf{W}(q) \otimes \mathbf{I}_d)\nu^{-1}(y + z)$. Kovalev et al. (2021) showed that multiplication with the gossip matrix can be seen as a compression on the Euclidean space $\mathcal{L}^{\perp}$, i.e., condition (3) holds, and hence the so-called error-feedback mechanism (Stich and Karimireddy, 2019; Karimireddy et al., 2019; Gorbunov et al., 2020) can be applied to obtain linear convergence. We use this insight in the design of our algorithm.

Armed with all these ideas, we are ready to present our method ADOM+; see Algorithm 1.

**Algorithm 1** ADOM+

1: **input:** $x^0, y^0, m^0 \in (\mathbb{R}^d)^{\mathcal{V}}, z^0 \in \mathcal{L}^{\perp}$
2: $x_f^0 = x^0, y_f^0 = y^0, z_f^0 = z^0$
3: **for** $k = 0, 1, 2, \ldots$ **do**
4:  $x_g^k = \tau_1 x^k + (1 - \tau_1) x_f^k$
5:  $x^{k+1} = x^k + \eta\alpha(x_g^k - x^{k+1}) - \eta \left[ \nabla F(x_g^k) - \nu x_g^k - y^{k+1} \right]$
6:  $x_f^{k+1} = x_g^k + \tau_2(x^{k+1} - x^k)$
7:  $y_g^k = \sigma_1 y^k + (1 - \sigma_1) y_f^k$
8:  $y^{k+1} = y^k + \theta\beta(\nabla F(x_g^k) - \nu x_g^k - y^{k+1}) - \theta \left[ \nu^{-1}(y_g^k + z_g^k) + x^{k+1} \right]$
9:  $y_f^{k+1} = y_g^k + \sigma_2(y^{k+1} - y^k)$
10: $z_g^k = \sigma_1 z^k + (1 - \sigma_1) z_f^k$
11: $z^{k+1} = z^k + \gamma\delta(z_g^k - z^k) - (\mathbf{W}(k) \otimes \mathbf{I}_d) \left[ \gamma\nu^{-1}(y_g^k + z_g^k) + m^k \right]$
12: $m^{k+1} = \gamma\nu^{-1}(y_g^k + z_g^k) + m^k - (\mathbf{W}(k) \otimes \mathbf{I}_d) \left[ \gamma\nu^{-1}(y_g^k + z_g^k) + m^k \right]$
13: $z_f^{k+1} = z_g^k - \zeta(\mathbf{W}(k) \otimes \mathbf{I}_d)(y_g^k + z_g^k)$
14: **end for**

We now establish the convergence rate of Algorithm 1.

**Theorem 4** (Convergence of ADOM+). *To reach precision $\|x^k - x^*\|^2 \leq \epsilon$, Algorithm 1 requires the following number of iterations*

$$\mathcal{O}\left( \chi\sqrt{L/\mu} \log \frac{1}{\epsilon} \right).$$

Note that Algorithm 1 performs $\mathcal{O}(1)$ decentralized communication and local computation rounds. Hence, its decentralized communication and local computation complexities are $\mathcal{O}\left( \chi\sqrt{L/\mu} \log \frac{1}{\epsilon} \right)$.

## 5 Optimal Algorithms

In this section we develop decentralized algorithms for time-varying networks with optimal local computation and decentralized communication complexities. We develop both primal and dual algorithms that use primal and dual gradient oracle, respectively. The key mechanism to obtain optimal algorithms is to incorporate the multi-consensus procedure into the algorithms with optimal decentralized communication complexity.

### 5.1 Multi-consensus Procedure

As discussed in Section 2.4, a decentralized communication round cab be represented as the multiplication with the gossip matrix $\mathbf{W}(q)$. The main idea behind the multi-consensus procedure is to replace the matrix $\mathbf{W}(q)$ with another matrix, namely

$$\mathbf{W}(k; T) = \mathbf{I}_n - \prod_{q=kT}^{(k+1)T-1} (\mathbf{I}_n - \mathbf{W}(q)), \tag{18}$$

where $k \in \{0, 1, 2, \ldots\}$ is the iteration counter, and $T \in \{1, 2, \ldots\}$ is the number of consensus steps. One can show that this matrix satisfies the Assumption 1, including the contraction property (3):

$$\|\mathbf{W}(k; T)x - x\|^2 \leq (1 - \chi^{-1})^T \|x\|^2 \text{ for all } x \in \mathcal{L}^{\perp}. \tag{19}$$

One can also observe that multiplication with the matrix $\mathbf{W}(k; T)$ requires to perform multiplication with $T$ gossip matrices $\mathbf{W}(kT), \mathbf{W}(kT+1), \ldots, \mathbf{W}(kT+T-1)$, hence it requires $T$ decentralized communication rounds.

### 5.2 Optimal Algorithms: ADOM and ADOM+ with multi-consensus

Now, we are ready to describe our optimal algorithms for smooth and strongly decentralized optimization over time-varying networks. As mentioned before, we start with algorithms with optimal

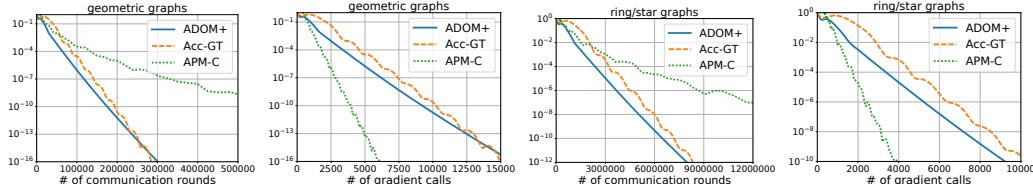

Figure 2: Comparison of our method ADOM+ and the baselines Acc-GT and APM-C.

decentralized communication complexity. In the case when access to the primal oracle is assumed, ADOM+ (Algorithm 1) is the algorithm of choice. In the case when access to the dual oracle is assumed, Kovalev et al. (2021) proposed the dual based accelerated decentralized algorithm ADOM. The original convergence proof of this algorithm requires the gossip matrix $\mathbf{W}(k)$ to be symmetric. However, the generalization of this proof to the case when the gossip matrix satisfies Assumption 1, and is not necessarily symmetric, is straightforward.

Both ADOM and ADOM+ require $\mathcal{O}\left(\chi\sqrt{L/\mu}\log\frac{1}{\epsilon}\right)$ iterations to obtain an $\epsilon$-accurate solution of problem (1), and at each iteration they perform $\mathcal{O}(1)$ local computation and decentralized communication rounds, i.e., multiplications with the gossip matrix $\mathbf{W}(q)$. We now incorporate a multi-consensus procedure into both algorithms. That is, we replace matrix $\mathbf{W}(q)$ with the matrix $\mathbf{W}(k;T)$ defined in (18), where $k \in \{0, 1, 2, \ldots\}$ is the iteration counter. As mentioned before, this is equivalent to performing $\mathcal{O}(T)$ decentralized communication rounds at each iteration. Choosing the number of consensus steps $T = \lceil \chi \ln 2 \rceil$ together with (19) implies

$$\|\mathbf{W}(k;T)x - x\|^2 \leq \tfrac{1}{2}\|x\|^2 \text{ for all } x \in \mathcal{L}^\perp. \tag{20}$$

This means that $\mathbf{W}(k;T)$ satisfies (3) with $\chi$ replaced by $\frac{1}{2}$ and hence both ADOM and ADOM+ with multi-consensus require $\mathcal{O}\left(\sqrt{L/\mu}\log\frac{1}{\epsilon}\right)$ iterations to obtain an $\epsilon$-accurate solution. Taking into account that these algorithms still perform $\mathcal{O}(1)$ local computations and $\mathcal{O}(T) = \mathcal{O}(\chi)$ decentralized communication rounds at each iteration, we arrive at the following theorems.

**Theorem 5.** *Let the functions $f_1, \ldots, f_n$ be $L$-smooth and $\mu$-strongly convex and let the Assumption 1 hold. Then, ADOM with multi-consensus requires $\mathcal{O}\left(\sqrt{L/\mu}\log\frac{1}{\epsilon}\right)$ local computation rounds and $\mathcal{O}\left(\chi\sqrt{L/\mu}\log\frac{1}{\epsilon}\right)$ decentralized communication rounds to find an $\epsilon$-accurate solution of the distributed optimization problem (1).*

**Theorem 6.** *Let the functions $f_1, \ldots, f_n$ be $L$-smooth and $\mu$-strongly convex and let the Assumption 1 hold. Then, ADOM+ with multi-consensus requires $\mathcal{O}\left(\sqrt{L/\mu}\log\frac{1}{\epsilon}\right)$ local computation rounds and $\mathcal{O}\left(\chi\sqrt{L/\mu}\log\frac{1}{\epsilon}\right)$ decentralized communication rounds to find an $\epsilon$-accurate solution of the distributed optimization problem (1).*

### 5.3 Comparison with Acc-GT (Li and Lin, 2021)

To the best of our knowledge ADOM with multi-consensus is the first dual based optimal decentralized algorithm for time-varying networks. In the case when the primal oracle is assumed, besides ADOM+ with multi-consensus, there is only one previously existing algorithm, Acc-GT with multi-consensus (Li and Lin, 2021), which achieves optimal local computation and decentralized communication complexities. However, in the case when multi-consensus is not used, its iteration complexity is $\mathcal{O}\left(\chi^{3/2}\sqrt{L/\mu}\log\frac{1}{\epsilon}\right)$, which is *worse* than the complexity of ADOM+, by the factor $\chi^{1/2}$. Since communication is known to be the main bottleneck in distributed training systems, multi-consensus is unlikely to be used in practice, which may limit the practical performance of Acc-GT in comparison to ADOM+.

## 6 Experiments

In this section we perform an illustrative experiment with logistic regression. We take $10,000$ samples from the `covtype` LIBSVM[6] dataset and distribute them across $n = 100$ nodes of a network, 100 samples per node. We use two types of networks: a sequence of random geometric graphs with $\chi \approx 30$, and a sequence which alternates between the ring and star topology, with $\chi \approx 1,000$. We choose the regularization parameter $\mu$ such that the condition number becomes $\kappa = 10^5$. We compare ADOM+ with multi-consensus with two state-of-the art decentralized algorithms for time-varying networks: Acc-GT with multi-consensus (Li and Lin, 2021), and a variant of APM-C for time-varying networks (Rogozin et al., 2020; Li et al., 2018). We set all parameters of Acc-GT and APM-C to those used in the experimental sections of the corresponding papers, and tune the parameters of ADOM+. The results are presented in Figure 2. We see that ADOM+ has similar empirical behavior to the recently proposed Acc-GT method. Both these methods are better than APM-C in terms of the number of decentralized communication rounds (this is expected, since this method has sublinear communication complexity), and worse in terms of the number of gradient calls. We provide more details and additional experiments in the appendix.

## Acknowledgments

The work of D. Kovalev, E. Gasanov and P. Richtárik was supported by the KAUST Baseline Research Funding Scheme. The work of A. Gasnikov was supported by the Russian Science Foundation (project 21-71-30005), a grant for research centers in the field of artificial intelligence, provided by the Analytical Center for the Government of the Russian Federation in accordance with the subsidy agreement (agreement identifier 000000D730321P5Q0002 ) and the agreement with the Ivannikov Institute for System Programming of the Russian Academy of Sciences dated November 2, 2021 No. 70-2021-00142.

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
