# Appendix: Lower Bounds and Optimal Algorithms. . .

## Contents

# A  Mixing and Gossip Matrices

In the literature on distributed optimization over time-varying networks mixing matrices are typically used to represent decentralized communication (Nedic et al., 2017). For each communication round $q \in \{0, 1, 2, \ldots\}$, consider a matrix $\mathbf{M}(q) \in \mathbb{R}^{n \times n}$ satisfying the following assumption.

**Assumption 2.** *For any $q \in \{0, 1, 2, \ldots\}$ matrix $\mathbf{M}(q) \in \mathbb{R}^{n \times n}$ satisfies the following relations:*

1. $\mathbf{M}(q)_{i,j} = 0$ *if $i \neq j$ and $(i, j) \notin \mathcal{E}^q$,*

2. $\mathbf{M}(q)\mathbf{1}_n = \mathbf{1}_n$ *and $\mathbf{1}_n^\top \mathbf{M}(q) = \mathbf{1}_n^\top$,*

3. *There exists $B \in \{1, 2, \ldots\}$ and $\chi \geq 1$ such that*

$$\sigma_{\max}^2 \left( \mathbf{M}_B(q) - \frac{1}{n}\mathbf{1}_n \mathbf{1}_n^\top \right) \leq (1 - \chi^{-1}),$$

*where $\sigma_{\max}(\cdot)$ denotes the largest singular value of a matrix and $\mathbf{M}_B(q)$ is defined as*

$$\mathbf{M}_B(q) = \prod_{q'=q}^{q+B-1} \mathbf{M}(q').$$

We will refer to the matrix $\mathbf{M}(q)$ as the *mixing matrix*. A typical example of the matrices $\mathbf{M}(q)$ is the Metropolis weight matrices for the sequence of $B$-connected graphs (Nedic et al., 2017).

Let us identify the difference between Assumption 2 on the mixing matrices $\mathbf{M}(q)$ and Assumption 1 on the gossip matrices. In order to do that define $\mathbf{W}(q) = \mathbf{I}_n - \mathbf{M}(q)$. In this case one can show that the matrices $\mathbf{M}(q)$ satisfy Assumption 2 if and only if the matrices $\mathbf{W}(q)$ satisfy Assumption 3, which is given as follow.

**Assumption 3.** *For any $q \in \{0, 1, 2, \ldots\}$ matrix $\mathbf{W}(q) \in \mathbb{R}^{n \times n}$ satisfies the following relations:*

1. $\mathbf{W}(q)_{i,j} = 0$ *if $i \neq j$ and $(i, j) \notin \mathcal{E}^q$,*

2. $\ker \mathbf{W}(q) \supset \{(x_1, \ldots, x_n) \in \mathbb{R}^n : x_1 = \ldots = x_n\}$,

3. $\mathrm{range}\mathbf{W}(q) \subset \{(x_1, \ldots, x_n) \in \mathbb{R}^n : \sum_{i=1}^n x_i = 0\}$,

4. *There exists $B \in \{1, 2, \ldots\}$ and $\chi \geq 1$ such that*

$$\|\mathbf{W}_B(q)x - x\|^2 \leq (1 - \chi^{-1})\|x\|^2 \text{ for all } x \in \left\{(x_1, \ldots, x_n) \in \mathbb{R}^n : \sum_{i=1}^n x_i = 0\right\},$$

*where $\mathbf{W}_B(q)$ is defined as*

$$\mathbf{W}_B(q) = \mathbf{I}_n - \prod_{q'=q}^{q+B-1} (\mathbf{I}_n - \mathbf{W}(q')).$$

Note that when $B = 1$, Assumption 3 coincides with Assumption 1. However, it turns out that our main results, including the lower bounds and optimal algorithms, can be easily extended to the general case $B > 1$.

## A.1  Lower Bounds

In this section we present Theorem 7, which establishes the lower bound on the number of decentralized communication rounds that are necessary to find an approximate solution of the problem (1) under Assumption 3. This is a generalized version of Theorem 2 for the case $B > 1$.

**Theorem 7.** *For any $\chi \geq 3$, $L > \mu \geq 0$ there exists a sequence of graphs $\{\mathcal{G}^q\}_{q=0}^\infty$, a corresponding sequence of gossip matrices $\{\hat{\mathbf{W}}(q)\}_{q=0}^\infty \subset \mathbb{R}^{n \times n}$ satisfying the Assumption 3, and $\mu$-strongly convex and $L$-smooth functions $f_i : \ell_2 \to \mathbb{R}, i \in \mathcal{V}$, such that for any first order decentralized algorithm, the number of decentralized communication rounds to find an $\epsilon$-accurate solution of the problem (1) is lower bounded by*

$$\Omega\left(B\chi\sqrt{L/\mu}\log\frac{1}{\epsilon}\right). \tag{21}$$

The proof is a trivial extension of the proof of Theorem 2. In particular, the only difference is that we use the sequence of the gossip matrices $\hat{\mathbf{W}}(q)$ which is given as

$$\hat{\mathbf{W}}(q) = \begin{cases} \mathbf{W}(q/B) & q \mod B = 0 \\ 0 & q \mod B \neq 0 \end{cases}, \tag{22}$$

where $\mathbf{W}(q)$ is the sequence of gossip matrices given by Theorem 2. Clearly, the matrices $\hat{\mathbf{W}}(q)$ satisfy Assumption 3. Such choice of the matrices $\hat{\mathbf{W}}(q)$ implies that nodes communicate only once per $B$ communication rounds. Hence, the lower complexity bound (21) given by Theorem 7 is $B$ times larger than the bound (5) given by Theorem 2.

## A.2 Optimal Algorithms

In this section we show how to develop optimal decentralized optimization algorithms for time-varying networks under the Assumption 3. It is done using the multi-consensus procedure similarly to Section 5. Indeed, let the matrices $\mathbf{W}(q)$ satisfy Assumption 3 and recall the matrix $\mathbf{W}(k;T)$ defined in (18). One can observe, that the matrices $\mathbf{W}(k;T)$, $k = 0, 1, 2 \ldots$, with $T = B$ satisfy our standard Assumption 1. Following Section 5.2 we choose $T = B\lceil \chi \ln 2 \rceil$ which implies (20). Hence, both ADOM and ADOM+ with $T = B\lceil \chi \ln 2 \rceil$ steps of multi-consensus achieve the iteration complexity $\mathcal{O}\left( \sqrt{L/\mu} \log \frac{1}{\epsilon} \right)$, which leads to the following theorems.

**Theorem 8.** *Let the functions $f_1, \ldots, f_n$ be L-smooth and $\mu$-strongly convex and let the Assumption 3 hold. Then, ADOM with multi-consensus requires $\mathcal{O}\left( \sqrt{L/\mu} \log \frac{1}{\epsilon} \right)$ local computation rounds and $\mathcal{O}\left( B\chi \sqrt{L/\mu} \log \frac{1}{\epsilon} \right)$ decentralized communication rounds to find an $\epsilon$-accurate solution of the distributed optimization problem* (1).

**Theorem 9.** *Let the functions $f_1, \ldots, f_n$ be L-smooth and $\mu$-strongly convex and let the Assumption 3 hold. Then, ADOM+ with multi-consensus requires $\mathcal{O}\left( \sqrt{L/\mu} \log \frac{1}{\epsilon} \right)$ local computation rounds and $\mathcal{O}\left( B\chi \sqrt{L/\mu} \log \frac{1}{\epsilon} \right)$ decentralized communication rounds to find an $\epsilon$-accurate solution of the distributed optimization problem* (1).

# B Experimental Details and Additional Experiments

## B.1 Experimental Details

We perform experiments with logistic regression for binary classification with $\ell_2$ regularization. That is, our loss function has the form

$$f_i(x) = \frac{1}{m} \sum_{j=1}^{m} \log(1 + \exp(-b_{ij} a_{ij}^\top x)) + \frac{r}{2}\|x\|^2, \tag{23}$$

where $a_{ij} \in \mathbb{R}^d$ and $b_{ij} \in \{-1, +1\}$ are data points and labels, $r > 0$ is a regularization parameter, and $m$ is the number of data points stored on each node. In this section we generate synthetic datasets with `sklearn.datasets.make classification` function from scikit-learn library. We generate a number of datasets consisting of $10,000$ samples, distributed across $n = 100$ nodes of the network, $m = 100$ samples per each node. We vary the parameter $r$ to obtain different values of the condition number $\kappa$.

## B.2 Further experiments

Here we simulate time-varying networks with a sequence of randomly generated geometric graphs. Geometric graphs are constructed by generating $n = 100$ nodes from the uniform distribution over $[0,1]^2 \subset \mathbb{R}^2$, and connecting each pair of nodes whose distance is less than a certain *radius*. We enforce connectivity by adding a small number of edges. We obtain a sequence of graphs $\{\mathcal{G}^q\}_{q=0}^\infty$ by generating a set of $50$ random geometric graphs, and alternating between them in a cyclic manner. We choose $\mathbf{W}(q)$ to be the Laplacian matrix of the graphs $\mathcal{G}^q$ divided by its largest eigenvalue. We vary the condition number $\chi$ by choosing different values of the radius parameter.

We compare ADOM+ with state-of-the art primal decentralized algorithms for time-varying networks: Acc-GT (Li and Lin, 2021) and a variant of APM-C (Rogozin et al., 2020; Li et al., 2018). We do not perform experiments with ADOM, because it is a dual based algorithm, and because its empirical behavior was studied in (Kovalev et al., 2021).

For each condition number of the problem $\kappa \in \{10, 10^2, 10^3, 10^4\}$, and condition number of the time-varying network $\chi \in \{3, 8, 37, 223, 2704, 4628\}$, we perform a comparison of these algorithms. Figures 3 and 4 show the convergence of the algorithms in the number of decentralized communications and the number of local computations, for all chosen values of $\kappa$ and $\chi$, respectively.

Overall, the results are similar to what was obtained in Section 6. We observe that ADOM+ and Acc-GT have similar behavior, which is expected since they are both optimal. Both of them perform better than APM-C in terms of the number of decentralized communication rounds, which is expected since APM-C has a sublinear communication complexity only. However, APM-C performs better in terms of the number of local computations. Indeed, while all three algorithms are optimal in local computation complexity,APM-C has better constants.

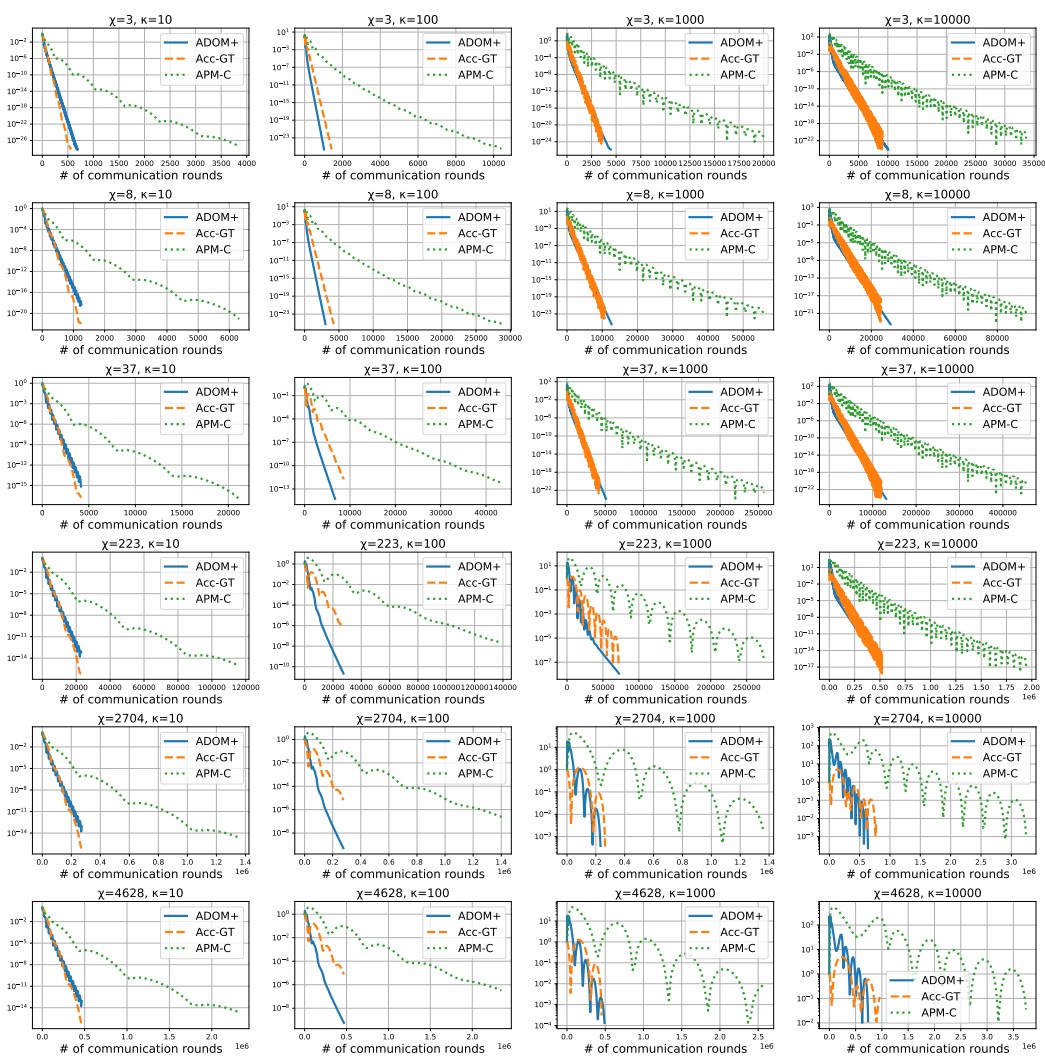

Figure 3: Comparison of our method ADOM+ and the baselines Acc-GT and APM-C in **decentralized communication complexity** on problems with $\kappa \in \{10, 10^2, 10^3, 10^4\}$ and time-varying networks with $\chi \in \{3, 8, 37, 223, 2704, 4628\}$.

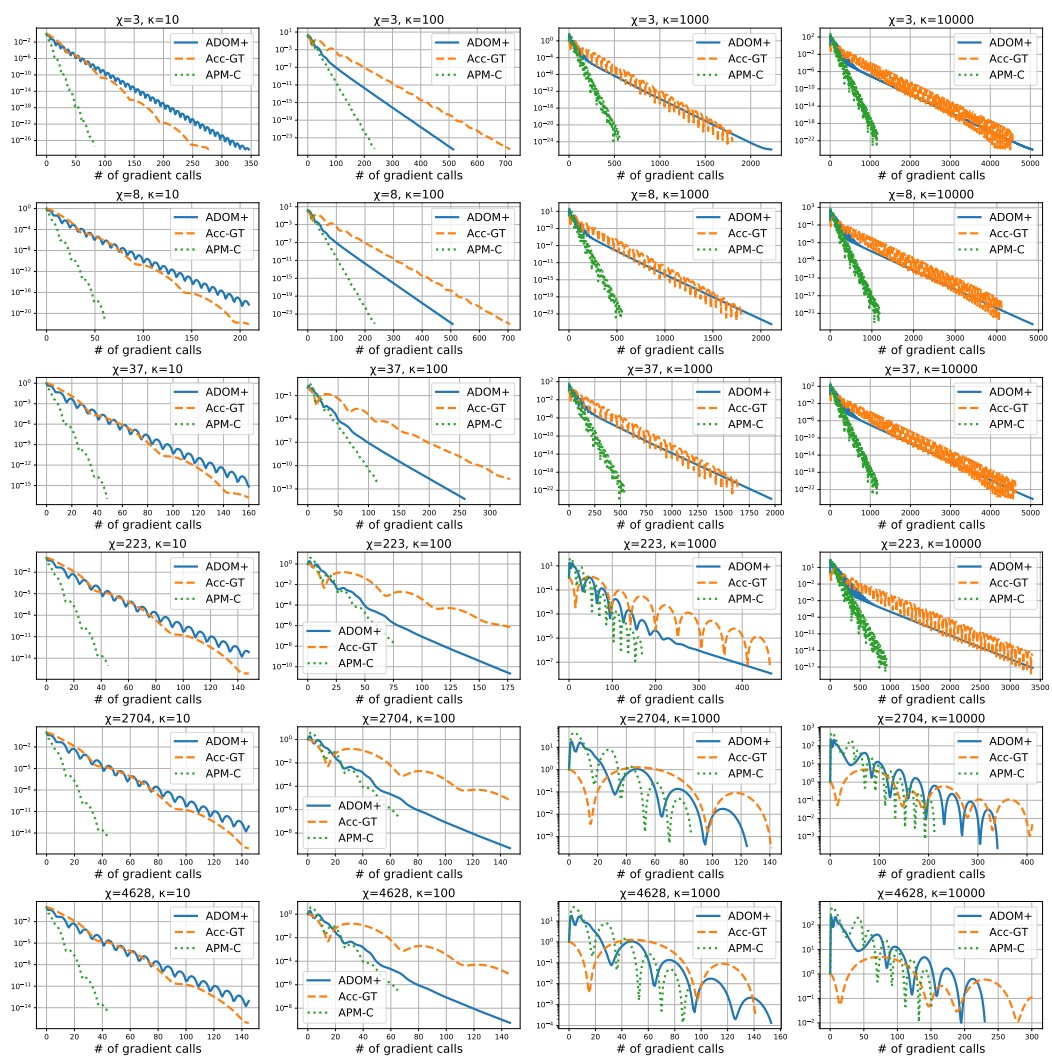

Figure 4: Comparison of our method ADOM+ and the baselines Acc-GT and APM-C in **local computation complexity** on problems with $\kappa \in \{10, 10^2, 10^3, 10^4\}$ and time-varying networks with $\chi \in \{3, 8, 37, 223, 2704, 4628\}$.

## C  Proof of Theorem 1

*Proof.* We choose number of nodes $n = 3\lfloor \chi/3 \rfloor$. Hence $n \geq 3$ and $n \mod 3 = 0$. Now, we divide the set of nodes $\mathcal{V} = \{1, \ldots, n\}$ into three disjoint sets $\mathcal{V} = \mathcal{V}_1 \cup \mathcal{V}_2 \cup \mathcal{V}_3$: $\mathcal{V}_1 = \{1, \ldots, n/3\}$, $\mathcal{V}_2 = \{n/3 + 1, \ldots, 2n/3\}$, $\mathcal{V}_3 = \{2n/3 + 1, \ldots, n\}$. Note, that $|\mathcal{V}_1| = |\mathcal{V}_2| = |\mathcal{V}_3| = n/3$.

We define $\mathcal{G}^q$ to be a star graph centered at the node $i_c(q) = |\mathcal{V}_1| + 1 + (q \mod |\mathcal{V}_2|)$. Note, that $i_c(q) \in \mathcal{V}_2$ for all $q \in \{0, 1, 2, \ldots\}$. We define $\mathbf{W}(q)$ as the Laplacian matrix of the graph $\mathcal{G}(q)$. Hence, $\lambda_{\max}(\mathbf{W}(q))/\lambda_{\min}^+(\mathbf{W}(q)) = n \leq \chi$ and $\mathbf{W}(q)$ satisfies condition (3).

We define functions $f_i$ in the following way:

$$
f_i(x) = \begin{cases} \frac{\mu}{2}\|x\|^2 + \frac{L-\mu}{4}\left[(x_{[1]} - 1)^2 + \sum_{l=1}^{\infty}(x_{[2l]} - x_{[2l+1]})^2\right], & i \in \mathcal{V}_1 \\ \frac{\mu}{2}\|x\|^2, & i \in \mathcal{V}_2 \\ \frac{\mu}{2}\|x\|^2 + \frac{L-\mu}{4}\sum_{l=1}^{\infty}(x_{[2l-1]} - x_{[2l]})^2, & i \in \mathcal{V}_3 \end{cases} \tag{24}
$$

The following lemma gives the solution of problem (1) with such a choice of $f_i$.

**Lemma 1.** *Problem* (1) *with* $f_i$ *given by* (24) *has a unique solution* $x^* = (\rho^l)_{l=1}^{\infty} \in \ell_2$, *where* $\rho$ *is given by*

$$
\rho = \frac{\sqrt{\frac{2L}{3\mu} + \frac{1}{3}} - 1}{\sqrt{\frac{2L}{3\mu} + \frac{1}{3}} + 1}. \tag{25}
$$

Consider the following quantity:

$$
s_i(k) = \begin{cases} 0, & \mathcal{H}_i(k) \subseteq \{0\}, \\ \min\{s \in \{1, 2, \ldots\} : \mathcal{H}_i(k) \subset \text{Span}(\{e_1, e_2, \ldots, e_s\})\}, & \text{otherwise} \end{cases}, \tag{26}
$$

where $e_s$ is the $s$-th unit basis vector. Using assumptions on the algorithm from Section 3.1 we can make some conclusions about update of $s_i(k)$. In particular, if at time step $k$ algorithm performs a local computation round, then using the fact, that $f_i(x)$ is a quadratic function with a certain block structure (and $f^*(x)$ has the same block structure), one can observe that

$$
s_i(k+1) \leq s_i(k) + \begin{cases} 1 - (s_i(k) \mod 2), & i \in \mathcal{V}_1 \\ 0, & i \in \mathcal{V}_2 \\ (s_i(k) \mod 2), & i \in \mathcal{V}_3 \end{cases}. \tag{27}
$$

Similarly, if at time step $k$ communication round number $q$ was performed, then using the structure of $\mathcal{G}^q$ one can observe that

$$
s_i(k+1) \leq \begin{cases} \max\{s_i(k), s_{i_c(q)}(k)\}, & i \neq i_c(q) \\ \max\{s_j(k) : j \in \mathcal{V}\}, & i = i_c(q) \end{cases}. \tag{28}
$$

The next key lemma shows that $s_i(k)$ is bounded compared to the number of communication rounds.

**Lemma 2.** *Let* $k \in \{0, 1, \ldots, k_q\}$ *be any time step, where* $k_q$ *is a time step at which the algorithm performed communication round number* $q \in \{0, 1, 2, \ldots\}$. *Then the following statement is true:*

$$
s_i(k) \leq 2\lfloor q/|\mathcal{V}_2| \rfloor + \begin{cases} 0, & i \in \mathcal{V}_3 \cup \{i_c(q), i_c(q) + 1, \ldots, |\mathcal{V}_1| + |\mathcal{V}_2|\} \\ 1, & \text{otherwise} \end{cases}. \tag{29}
$$

Lemma 2 implies

$$
s_i(k) \leq \frac{2q}{|\mathcal{V}_2|} + 1 = \frac{6q}{n} + 1 = \frac{2q}{\lfloor \chi/3 \rfloor} + 1 \leq \frac{12q}{\chi} + 1.
$$

Hence, using Lemma (1) we can lower bound $\|x_i^o(k) - x^*\|^2$:

$$
\|x_i^o(k) - x^*\|^2 = \sum_{l=1}^{\infty}(x_i^o(k) - x^*)_{[l]}^2 \geq \sum_{l=s_i(k)+1}^{\infty}(x_i^o(k) - x^*)_{[l]}^2
$$

$$= \sum_{l=s_i(k)+1}^{\infty} \rho^{2l} = \rho^{2s_i(k)+2} \sum_{l=0}^{\infty} \rho^{2l} = \frac{\rho^{2s_i(k)+2}}{1-\rho^2}$$

$$\geq \frac{\rho^{24q/\chi+4}}{1-\rho^2} \geq C\rho^{24q/\chi},$$

where $C = \frac{\rho^4}{1-\rho^2}$. Note, that $\rho \geq \max\left\{0, 1 - \frac{\sqrt{6\mu}}{\sqrt{L}}\right\}$ and hence

$$\|x_i^o(k) - x^*\|^2 \geq C\left(\max\left\{0, 1 - \frac{\sqrt{6\mu}}{\sqrt{L}}\right\}\right)^{24q/\chi}.$$

Finally, using Bernoulli inequality we get

$$\|x_i^o(k) - x^*\|^2 \geq C\left(\max\left\{0, 1 - \frac{24\sqrt{6\mu}}{\sqrt{L}}\right\}\right)^{q/\chi}.$$

$\square$

*Proof of Lemma 1.*

$$\sum_{i \in \mathcal{V}} f_i(x) = \frac{n\mu}{2}\|x\|^2 + \frac{n(L-\mu)}{12}\left[(x_{[1]}-1)^2 + \sum_{l=1}^{\infty}(x_{[l]} - x_{[l+1]})^2\right]$$

$$= \frac{n\mu}{2}\|x\|^2 + \frac{n(L-\mu)}{12}\left[x_{[1]}^2 - 2x_{[1]} + 1 + \sum_{l=1}^{\infty}(x_{[l]}^2 - 2x_{[l]}x_{[l+1]} + x_{[l+1]}^2)\right]$$

$$= \frac{n\mu}{2}\|x\|^2 + \frac{n(L-\mu)}{12}\left[\sum_{l=1}^{\infty}(2x_{[l]}^2 - 2x_{[l]}x_{[l+1]}) - 2x_{[1]} + 1\right]$$

$$= \frac{n(L-\mu)}{12}\left[\sum_{l=1}^{\infty}\left(\left(2 + \frac{6\mu}{L-\mu}\right)x_{[l]}^2 - 2x_{[l]}x_{[l+1]}\right) - 2x_{[1]} + 1\right]$$

$$= \frac{n(L-\mu)}{12}\left[\sum_{l=1}^{\infty}\left(\frac{2(L+2\mu)}{L-\mu}x_{[l]}^2 - 2x_{[l]}x_{[l+1]}\right) - 2x_{[1]} + 1\right].$$

Next, we use the fact that $\frac{2(L+2\mu)}{L-\mu} = \rho + \frac{1}{\rho}$ with $\rho$ given by (25) and get

$$\sum_{i \in \mathcal{V}} f_i(x) = \frac{n(L-\mu)}{12}\left[\sum_{l=1}^{\infty}\left(\rho x_{[l]}^2 + \frac{1}{\rho}x_{[l]}^2 - 2x_{[l]}x_{[l+1]}\right) - 2x_{[1]} + 1\right]$$

$$= \frac{n(L-\mu)}{12\rho}\left[\sum_{l=1}^{\infty}\left(\rho^2 x_{[l]}^2 + x_{[l]}^2 - 2\rho x_{[l]}x_{[l+1]}\right) - 2\rho x_{[1]} + \rho\right]$$

$$= \frac{n(L-\mu)}{12\rho}\left[\sum_{l=1}^{\infty}\left(\rho^2 x_{[l]}^2 - 2\rho x_{[l]}x_{[l+1]} + x_{[l+1]}^2\right) + x_{[1]}^2 - 2\rho x_{[1]} + \rho\right]$$

$$= \frac{n(L-\mu)}{12\rho}\left[\sum_{l=1}^{\infty}(\rho x_{[l]} - x_{[l+1]})^2 + (x_{[1]} - \rho)^2 + \rho - \rho^2\right].$$

Now it's clear that $\sum_{i \in \mathcal{V}} f_i(x) \geq \frac{n(L-\mu)(1-\rho)}{12}$ and $\sum_{i \in \mathcal{V}} f_i(x) = \frac{n(L-\mu)(1-\rho)}{12}$ if and only if $x = x^*$. Hence, $x^*$ is indeed a unique solution to the problem (1). $\square$

*Proof of Lemma 2.* We prove this by induction in $q$.

**Induction basis.** When $q = 0$ and $k \in \{0, 1, \ldots, k_0\}$ (meaning no communication rounds were done), from (27) we can conclude, that

$$s_i(k) \leq \begin{cases} 1, & i \in \mathcal{V}_1 \\ 0, & i \in \mathcal{V}_2 \cup \mathcal{V}_3 \end{cases},$$

which is (29) in case $q = 0$.

**Induction step.** Now, we assume that (29) holds for $q$ and $k \in \{0, 1, \ldots, k_q\}$ and prove it for $q + 1$ and $k \in \{0, 1, \ldots, k_{q+1}\}$. Indeed, consider time step $k_q$ at which communication round $q$ is performed. Consider two possible cases:

1. $i_c(q) \neq |\mathcal{V}_1| + |\mathcal{V}_2|$.
   In this case $\lfloor (q+1)/\mathcal{V}_2 \rfloor = \lfloor q/\mathcal{V}_2 \rfloor$ and $i_c(q+1) = i_c(q) + 1$. Since (29) holds for $q$ and $k_q$, using (28) we get $s_{i_c(q)}(k_q) \leq 2\lfloor q/\mathcal{V}_2 \rfloor$ and $s_{i_c(q)}(k_q + 1) \leq 2\lfloor q/\mathcal{V}_2 \rfloor + 1$. Hence, using (28) we get for all $i \in \mathcal{V}$

$$s_i(k) \leq 2\lfloor q/|\mathcal{V}_2| \rfloor + \begin{cases} 0, & i \in \mathcal{V}_3 \cup \{i_c(q) + 1, \ldots, |\mathcal{V}_1| + |\mathcal{V}_2|\} \\ 1, & \text{otherwise} \end{cases}$$

$$= 2\lfloor (q+1)/|\mathcal{V}_2| \rfloor + \begin{cases} 0, & i \in \mathcal{V}_3 \cup \{i_c(q) + 1, \ldots, |\mathcal{V}_1| + |\mathcal{V}_2|\} \\ 1, & \text{otherwise} \end{cases},$$

   which is (29) for $q + 1$ and $k = k_q + 1$.

2. $i_c(q) = |\mathcal{V}_1| + |\mathcal{V}_2|$.
   In this case $\lfloor (q+1)/\mathcal{V}_2 \rfloor = \lfloor q/\mathcal{V}_2 \rfloor + 1$ and $i_c(q+1) = |\mathcal{V}_1| + 1$. Since (29) holds for $q$ and $k_q$, we have an upper bound $s_i(k_q) \leq 2\lfloor q/\mathcal{V}_2 \rfloor + 1$ and (28) implies

$$s_i(k_q + 1) \leq 2\lfloor q/\mathcal{V}_2 \rfloor + 1 \leq 2(\lfloor q/\mathcal{V}_2 \rfloor + 1) = 2\lfloor (q+1)/\mathcal{V}_2 \rfloor$$

$$\leq 2\lfloor (q+1)/|\mathcal{V}_2| \rfloor + \begin{cases} 0, & i \in \mathcal{V}_3 \cup \{i_c(q+1), \ldots, |\mathcal{V}_1| + |\mathcal{V}_2|\} \\ 1, & \text{otherwise} \end{cases}.$$

   which is (29) for $q + 1$ and $k = k_q + 1$.

In both cases we have obtained (29) for $q + 1$ and $k = k_q + 1$. It remains to see that at any time step $k \in \{k_q + 2, k_{q+1} - 1\}$ only local computation is performed and (27) implies that (29) also holds for $q + 1$ and any $k \in \{0, 1, \ldots, k_{q+1}\}$.

$\square$

# D  Proof of Theorem 4

By $\mathrm{D}_F(x, y)$ we denote Bregman distance $\mathrm{D}_F(x, y) := F(x) - F(y) - \langle \nabla F(y), x - y \rangle$.

**Lemma 3.** *Let $\tau_2$ be defined as follows:*

$$\tau_2 = \sqrt{\mu/L}. \tag{30}$$

*Let $\tau_1$ be defined as follows:*

$$\tau_1 = (1/\tau_2 + 1/2)^{-1}. \tag{31}$$

*Let $\eta$ be defined as follows:*

$$\eta = (L\tau_2)^{-1}. \tag{32}$$

*Let $\alpha$ be defined as follows:*

$$\alpha = \mu/2. \tag{33}$$

*Let $\nu$ be defined as follows:*

$$\nu = \mu/2. \tag{34}$$

*Let $\Psi_x^k$ be defined as follows:*

$$\Psi_x^k = \left( \frac{1}{\eta} + \alpha \right) \|x^k - x^*\|^2 + \frac{2}{\tau_2} \left( \mathrm{D}_f(x_f^k, x^*) - \frac{\nu}{2} \|x_f^k - x^*\|^2 \right) \tag{35}$$

*Then the following inequality holds:*

$$\Psi_x^{k+1} \leq \left( 1 - \frac{\sqrt{\mu}}{\sqrt{\mu} + 2\sqrt{L}} \right) \Psi_x^k + 2\langle y^{k+1} - y^*, x^{k+1} - x^* \rangle - \left( \mathrm{D}_F(x_g^k, x^*) - \frac{\nu}{2} \|x_g^k - x^*\|^2 \right). \tag{36}$$

*Proof.*

$$\frac{1}{\eta} \|x^{k+1} - x^*\|^2 = \frac{1}{\eta} \|x^k - x^*\|^2 + \frac{2}{\eta} \langle x^{k+1} - x^k, x^{k+1} - x^* \rangle - \frac{1}{\eta} \|x^{k+1} - x^k\|^2.$$

Using Line 5 of Algorithm 1 we get

$$\frac{1}{\eta} \|x^{k+1} - x^*\|^2 = \frac{1}{\eta} \|x^k - x^*\|^2 + 2\alpha \langle x_g^k - x^{k+1}, x^{k+1} - x^* \rangle$$

$$- 2\langle \nabla F(x_g^k) - \nu x_g^k - y^{k+1}, x^{k+1} - x^* \rangle - \frac{1}{\eta} \|x^{k+1} - x^k\|^2$$

$$= \frac{1}{\eta} \|x^k - x^*\|^2 + 2\alpha \langle x_g^k - x^* - x^{k+1} + x^*, x^{k+1} - x^* \rangle$$

$$- 2\langle \nabla F(x_g^k) - \nu x_g^k - y^{k+1}, x^{k+1} - x^* \rangle - \frac{1}{\eta} \|x^{k+1} - x^k\|^2$$

$$\leq \frac{1}{\eta} \|x^k - x^*\|^2 - \alpha \|x^{k+1} - x^*\|^2 + \alpha \|x_g^k - x^*\|^2 - 2\langle \nabla F(x_g^k) - \nu x_g^k - y^{k+1}, x^{k+1} - x^* \rangle$$

$$- \frac{1}{\eta} \|x^{k+1} - x^k\|^2.$$

Using optimality condition (12) we get

$$\frac{1}{\eta} \|x^{k+1} - x^*\|^2 \leq \frac{1}{\eta} \|x^k - x^*\|^2 - \alpha \|x^{k+1} - x^*\|^2 + \alpha \|x_g^k - x^*\|^2 - \frac{1}{\eta} \|x^{k+1} - x^k\|^2$$

$$- 2\langle \nabla F(x_g^k) - \nabla F(x^*), x^{k+1} - x^* \rangle + 2\nu \langle x_g^k - x^*, x^{k+1} - x^* \rangle + 2\langle y^{k+1} - y^*, x^{k+1} - x^* \rangle.$$

Using Line 6 of Algorithm 1 we get

$$\frac{1}{\eta} \|x^{k+1} - x^*\|^2 \leq \frac{1}{\eta} \|x^k - x^*\|^2 - \alpha \|x^{k+1} - x^*\|^2 + \alpha \|x_g^k - x^*\|^2 - \frac{1}{\eta \tau_2^2} \|x_f^{k+1} - x_g^k\|^2$$

$$- 2\langle \nabla F(x_g^k) - \nabla F(x^*), x^k - x^* \rangle + 2\nu \langle x_g^k - x^*, x^k - x^* \rangle + 2\langle y^{k+1} - y^*, x^{k+1} - x^* \rangle$$

$$
- \frac{2}{\tau_2} \langle \nabla F(x_g^k) - \nabla F(x^*), x_f^{k+1} - x_g^k \rangle + \frac{2\nu}{\tau_2} \langle x_g^k - x^*, x_f^{k+1} - x_g^k \rangle
$$

$$
= \frac{1}{\eta} \| x^k - x^* \|^2 - \alpha \| x^{k+1} - x^* \|^2 + \alpha \| x_g^k - x^* \|^2 - \frac{1}{\eta \tau_2^2} \| x_f^{k+1} - x_g^k \|^2
$$
$$
- 2 \langle \nabla F(x_g^k) - \nabla F(x^*), x^k - x^* \rangle + 2\nu \langle x_g^k - x^*, x^k - x^* \rangle + 2 \langle y^{k+1} - y^*, x^{k+1} - x^* \rangle
$$
$$
- \frac{2}{\tau_2} \langle \nabla F(x_g^k) - \nabla F(x^*), x_f^{k+1} - x_g^k \rangle + \frac{\nu}{\tau_2} \left( \| x_f^{k+1} - x^* \|^2 - \| x_g^k - x^* \|^2 - \| x_f^{k+1} - x_g^k \|^2 \right).
$$

Using $L$-smoothness of $\mathrm{D}_F(x, x^*)$ in $x$, which follows from $L$-smoothness of $F(x)$, we get

$$
\frac{1}{\eta} \| x^{k+1} - x^* \|^2 \le \frac{1}{\eta} \| x^k - x^* \|^2 - \alpha \| x^{k+1} - x^* \|^2 + \alpha \| x_g^k - x^* \|^2 - \frac{1}{\eta \tau_2^2} \| x_f^{k+1} - x_g^k \|^2
$$
$$
- 2 \langle \nabla F(x_g^k) - \nabla F(x^*), x^k - x^* \rangle + 2\nu \langle x_g^k - x^*, x^k - x^* \rangle + 2 \langle y^{k+1} - y^*, x^{k+1} - x^* \rangle
$$
$$
- \frac{2}{\tau_2} \langle \nabla F(x_g^k) - \nabla F(x^*), x_f^{k+1} - x_g^k \rangle + \frac{\nu}{\tau_2} \left( \| x_f^{k+1} - x^* \|^2 - \| x_g^k - x^* \|^2 - \| x_f^{k+1} - x_g^k \|^2 \right)
$$
$$
\le \frac{1}{\eta} \| x^k - x^* \|^2 - \alpha \| x^{k+1} - x^* \|^2 + \alpha \| x_g^k - x^* \|^2 - \frac{1}{\eta \tau_2^2} \| x_f^{k+1} - x_g^k \|^2
$$
$$
- 2 \langle \nabla F(x_g^k) - \nabla F(x^*), x^k - x^* \rangle + 2\nu \langle x_g^k - x^*, x^k - x^* \rangle + 2 \langle y^{k+1} - y^*, x^{k+1} - x^* \rangle
$$
$$
- \frac{2}{\tau_2} \left( \mathrm{D}_f(x_f^{k+1}, x^*) - \mathrm{D}_f(x_g^k, x^*) - \frac{L}{2} \| x_f^{k+1} - x_g^k \|^2 \right)
$$
$$
+ \frac{\nu}{\tau_2} \left( \| x_f^{k+1} - x^* \|^2 - \| x_g^k - x^* \|^2 - \| x_f^{k+1} - x_g^k \|^2 \right)
$$
$$
= \frac{1}{\eta} \| x^k - x^* \|^2 - \alpha \| x^{k+1} - x^* \|^2 + \alpha \| x_g^k - x^* \|^2 + \left( \frac{L - \nu}{\tau_2} - \frac{1}{\eta \tau_2^2} \right) \| x_f^{k+1} - x_g^k \|^2
$$
$$
- 2 \langle \nabla F(x_g^k) - \nabla F(x^*), x^k - x^* \rangle + 2\nu \langle x_g^k - x^*, x^k - x^* \rangle + 2 \langle y^{k+1} - y^*, x^{k+1} - x^* \rangle
$$
$$
- \frac{2}{\tau_2} \left( \mathrm{D}_f(x_f^{k+1}, x^*) - \mathrm{D}_f(x_g^k, x^*) \right) + \frac{\nu}{\tau_2} \left( \| x_f^{k+1} - x^* \|^2 - \| x_g^k - x^* \|^2 \right)
$$

Using Line 4 of Algorithm 1 we get

$$
\frac{1}{\eta} \| x^{k+1} - x^* \|^2 \le \frac{1}{\eta} \| x^k - x^* \|^2 - \alpha \| x^{k+1} - x^* \|^2 + \alpha \| x_g^k - x^* \|^2 + \left( \frac{L - \nu}{\tau_2} - \frac{1}{\eta \tau_2^2} \right) \| x_f^{k+1} - x_g^k \|^2
$$
$$
- 2 \langle \nabla F(x_g^k) - \nabla F(x^*), x_g^k - x^* \rangle + 2\nu \| x_g^k - x^* \|^2 + \frac{2(1 - \tau_1)}{\tau_1} \langle \nabla F(x_g^k) - \nabla F(x^*), x_f^k - x_g^k \rangle
$$
$$
+ \frac{2\nu(1 - \tau_1)}{\tau_1} \langle x_g^k - x_f^k, x_g^k - x^* \rangle + 2 \langle y^{k+1} - y^*, x^{k+1} - x^* \rangle
$$
$$
- \frac{2}{\tau_2} \left( \mathrm{D}_f(x_f^{k+1}, x^*) - \mathrm{D}_f(x_g^k, x^*) \right) + \frac{\nu}{\tau_2} \left( \| x_f^{k+1} - x^* \|^2 - \| x_g^k - x^* \|^2 \right)
$$
$$
= \frac{1}{\eta} \| x^k - x^* \|^2 - \alpha \| x^{k+1} - x^* \|^2 + \alpha \| x_g^k - x^* \|^2 + \left( \frac{L - \nu}{\tau_2} - \frac{1}{\eta \tau_2^2} \right) \| x_f^{k+1} - x_g^k \|^2
$$
$$
- 2 \langle \nabla F(x_g^k) - \nabla F(x^*), x_g^k - x^* \rangle + 2\nu \| x_g^k - x^* \|^2 + \frac{2(1 - \tau_1)}{\tau_1} \langle \nabla F(x_g^k) - \nabla F(x^*), x_f^k - x_g^k \rangle
$$
$$
+ \frac{\nu(1 - \tau_1)}{\tau_1} \left( \| x_g^k - x_f^k \|^2 + \| x_g^k - x^* \|^2 - \| x_f^k - x^* \|^2 \right) + 2 \langle y^{k+1} - y^*, x^{k+1} - x^* \rangle
$$
$$
- \frac{2}{\tau_2} \left( \mathrm{D}_f(x_f^{k+1}, x^*) - \mathrm{D}_f(x_g^k, x^*) \right) + \frac{\nu}{\tau_2} \left( \| x_f^{k+1} - x^* \|^2 - \| x_g^k - x^* \|^2 \right).
$$

Using $\mu$-strong convexity of $\mathrm{D}_F(x, x^*)$ in $x$, which follows from $\mu$-strong convexity of $F(x)$, we get

$$
\frac{1}{\eta} \| x^{k+1} - x^* \|^2 \le \frac{1}{\eta} \| x^k - x^* \|^2 - \alpha \| x^{k+1} - x^* \|^2 + \alpha \| x_g^k - x^* \|^2 + \left( \frac{L - \nu}{\tau_2} - \frac{1}{\eta \tau_2^2} \right) \| x_f^{k+1} - x_g^k \|^2
$$
$$
- 2 \mathrm{D}_F(x_g^k, x^*) - \mu \| x_g^k - x^* \|^2 + 2\nu \| x_g^k - x^* \|^2
$$

$$+ \frac{2(1-\tau_1)}{\tau_1}\left(D_F(x_f^k, x^*) - D_F(x_g^k, x^*) - \frac{\mu}{2}\|x_f^k - x_g^k\|^2\right)$$

$$+ \frac{\nu(1-\tau_1)}{\tau_1}\left(\|x_g^k - x_f^k\|^2 + \|x_g^k - x^*\|^2 - \|x_f^k - x^*\|^2\right) + 2\langle y^{k+1} - y^*, x^{k+1} - x^*\rangle$$

$$- \frac{2}{\tau_2}\left(D_f(x_f^{k+1}, x^*) - D_f(x_g^k, x^*)\right) + \frac{\nu}{\tau_2}\left(\|x_f^{k+1} - x^*\|^2 - \|x_g^k - x^*\|^2\right)$$

$$= \frac{1}{\eta}\|x^k - x^*\|^2 - \alpha\|x^{k+1} - x^*\|^2 + \frac{2(1-\tau_1)}{\tau_1}\left(D_F(x_f^k, x^*) - \frac{\nu}{2}\|x_f^k - x^*\|^2\right)$$

$$- \frac{2}{\tau_2}\left(D_f(x_f^{k+1}, x^*) - \frac{\nu}{2}\|x_f^{k+1} - x^*\|^2\right) + 2\langle y^{k+1} - y^*, x^{k+1} - x^*\rangle$$

$$+ 2\left(\frac{1}{\tau_2} - \frac{1}{\tau_1}\right)D_F(x_g^k, x^*) + \left(\alpha - \mu + \nu + \frac{\nu}{\tau_1} - \frac{\nu}{\tau_2}\right)\|x_g^k - x^*\|^2$$

$$+ \left(\frac{L-\nu}{\tau_2} - \frac{1}{\eta\tau_2^2}\right)\|x_f^{k+1} - x_g^k\|^2 + \frac{(1-\tau_1)(\nu-\mu)}{\tau_1}\|x_f^k - x_g^k\|^2.$$

Using $\eta$ defined by (32), $\tau_1$ defined by (31) and the fact that $\nu < \mu$ we get

$$\frac{1}{\eta}\|x^{k+1} - x^*\|^2 \le \frac{1}{\eta}\|x^k - x^*\|^2 - \alpha\|x^{k+1} - x^*\|^2 + \frac{2(1-\tau_2/2)}{\tau_2}\left(D_F(x_f^k, x^*) - \frac{\nu}{2}\|x_f^k - x^*\|^2\right)$$

$$- \frac{2}{\tau_2}\left(D_f(x_f^{k+1}, x^*) - \frac{\nu}{2}\|x_f^{k+1} - x^*\|^2\right) + 2\langle y^{k+1} - y^*, x^{k+1} - x^*\rangle$$

$$- D_F(x_g^k, x^*) + \left(\alpha - \mu + \frac{3\nu}{2}\right)\|x_g^k - x^*\|^2.$$

Using $\alpha$ defined by (33) and $\nu$ defined by (34) we get

$$\frac{1}{\eta}\|x^{k+1} - x^*\|^2 \le \frac{1}{\eta}\|x^k - x^*\|^2 - \alpha\|x^{k+1} - x^*\|^2 + \frac{2(1-\tau_2/2)}{\tau_2}\left(D_F(x_f^k, x^*) - \frac{\nu}{2}\|x_f^k - x^*\|^2\right)$$

$$- \frac{2}{\tau_2}\left(D_f(x_f^{k+1}, x^*) - \frac{\nu}{2}\|x_f^{k+1} - x^*\|^2\right) + 2\langle y^{k+1} - y^*, x^{k+1} - x^*\rangle$$

$$- \left(D_F(x_g^k, x^*) - \frac{\nu}{2}\|x_g^k - x^*\|^2\right).$$

After rearranging and using $\Psi_x^k$ definition (35) we get

$$\Psi_x^{k+1} \le \max\left\{1 - \tau_2/2, 1/(1+\eta\alpha)\right\}\Psi_x^k + 2\langle y^{k+1} - y^*, x^{k+1} - x^*\rangle - \left(D_F(x_g^k, x^*) - \frac{\nu}{2}\|x_g^k - x^*\|^2\right)$$

$$\le \left(1 - \frac{\sqrt{\mu}}{\sqrt{\mu} + 2\sqrt{L}}\right)\Psi_x^k + 2\langle y^{k+1} - y^*, x^{k+1} - x^*\rangle - \left(D_F(x_g^k, x^*) - \frac{\nu}{2}\|x_g^k - x^*\|^2\right).$$

$\square$

**Lemma 4.** *The following inequality holds:*

$$-\|y^{k+1} - y^*\|^2 \le \frac{(1-\sigma_1)}{\sigma_1}\|y_f^k - y^*\|^2 - \frac{1}{\sigma_2}\|y_f^{k+1} - y^*\|^2$$
$$- \left(\frac{1}{\sigma_1} - \frac{1}{\sigma_2}\right)\|y_g^k - y^*\|^2 + (\sigma_2 - \sigma_1)\|y^{k+1} - y^k\|^2$$

(37)

*Proof.* Lines 7 and 9 of Algorithm 1 imply

$$y_f^{k+1} = y_g^k + \sigma_2(y^{k+1} - y_k)$$
$$= y_g^k + \sigma_2 y^{k+1} - \frac{\sigma_2}{\sigma_1}\left(y_g^k - (1-\sigma_1)y_f^k\right)$$
$$= \left(1 - \frac{\sigma_2}{\sigma_1}\right)y_g^k + \sigma_2 y^{k+1} + \left(\frac{\sigma_2}{\sigma_1} - \sigma_2\right)y_f^k.$$

After subtracting $y^*$ and rearranging we get

$$(y_f^{k+1} - y^*) + \left(\frac{\sigma_2}{\sigma_1} - 1\right)(y_g^k - y^*) = \sigma_2(y^{k+1} - y^*) + \left(\frac{\sigma_2}{\sigma_1} - \sigma_2\right)(y_f^k - y^*).$$

Multiplying both sides by $\frac{\sigma_1}{\sigma_2}$ gives

$$\frac{\sigma_1}{\sigma_2}(y_f^{k+1} - y^*) + \left(1 - \frac{\sigma_1}{\sigma_2}\right)(y_g^k - y^*) = \sigma_1(y^{k+1} - y^*) + (1 - \sigma_1)(y_f^k - y^*).$$

Squaring both sides gives

$$\frac{\sigma_1}{\sigma_2}\|y_f^{k+1} - y^*\|^2 + \left(1 - \frac{\sigma_1}{\sigma_2}\right)\|y_g^k - y^*\|^2 - \frac{\sigma_1}{\sigma_2}\left(1 - \frac{\sigma_1}{\sigma_2}\right)\|y_f^{k+1} - y_g^k\|^2 \leq \sigma_1\|y^{k+1} - y^*\|^2 + (1 - \sigma_1)\|y_f^k - y^*\|^2.$$

Rearranging gives

$$-\|y^{k+1} - y^*\|^2 \leq -\left(\frac{1}{\sigma_1} - \frac{1}{\sigma_2}\right)\|y_g^k - y^*\|^2 + \frac{(1 - \sigma_1)}{\sigma_1}\|y_f^k - y^*\|^2 - \frac{1}{\sigma_2}\|y_f^{k+1} - y^*\|^2 + \frac{1}{\sigma_2}\left(1 - \frac{\sigma_1}{\sigma_2}\right)\|y_f^{k+1} - y_g^k\|^2.$$

Using Line 9 of Algorithm 1 we get

$$-\|y^{k+1} - y^*\|^2 \leq -\left(\frac{1}{\sigma_1} - \frac{1}{\sigma_2}\right)\|y_g^k - y^*\|^2 + \frac{(1 - \sigma_1)}{\sigma_1}\|y_f^k - y^*\|^2 - \frac{1}{\sigma_2}\|y_f^{k+1} - y^*\|^2 + (\sigma_2 - \sigma_1)\|y^{k+1} - y^k\|^2.$$

$\square$

**Lemma 5.** *Let $\beta$ be defined as follows:*

$$\beta = 1/(2L). \tag{38}$$

*Let $\sigma_1$ be defined as follows:*

$$\sigma_1 = (1/\sigma_2 + 1/2)^{-1}. \tag{39}$$

*Then the following inequality holds:*

$$\left(\frac{1}{\theta} + \frac{\beta}{2}\right)\|y^{k+1} - y^*\|^2 + \frac{\beta}{2\sigma_2}\|y_f^{k+1} - y^*\|^2$$

$$\leq \frac{1}{\theta}\|y^k - y^*\|^2 + \frac{\beta(1 - \sigma_2/2)}{2\sigma_2}\|y_f^k - y^*\|^2 + \mathrm{D}_F(x_g^k, x^*) - \frac{\nu}{2}\|x_g^k - x^*\|^2 - 2\langle x^{k+1} - x^*, y^{k+1} - y^*\rangle$$

$$- 2\nu^{-1}\langle y_g^k + z_g^k - (y^* + z^*), y^{k+1} - y^*\rangle - \frac{\beta}{4}\|y_g^k - y^*\|^2 + \left(\frac{\beta\sigma_2^2}{4} - \frac{1}{\theta}\right)\|y^{k+1} - y^k\|^2. \tag{40}$$

*Proof.*

$$\frac{1}{\theta}\|y^{k+1} - y^*\|^2 = \frac{1}{\theta}\|y^k - y^*\|^2 + \frac{2}{\theta}\langle x^{k+1} - x^k, x^{k+1} - x^*\rangle - \frac{1}{\theta}\|y^{k+1} - y^k\|^2.$$

Using Line 8 of Algorithm 1 we get

$$\frac{1}{\theta}\|y^{k+1} - y^*\|^2 = \frac{1}{\theta}\|y^k - y^*\|^2 + 2\beta\langle \nabla F(x_g^k) - \nu x_g^k - y^{k+1}, y^{k+1} - y^*\rangle$$

$$- 2\langle \nu^{-1}(y_g^k + z_g^k) + x^{k+1}, y^{k+1} - y^*\rangle - \frac{1}{\theta}\|y^{k+1} - y^k\|^2.$$

Using optimality condition (12) we get

$$\frac{1}{\theta}\|y^{k+1} - y^*\|^2 = \frac{1}{\theta}\|y^k - y^*\|^2 + 2\beta\langle \nabla F(x_g^k) - \nu x_g^k - (\nabla F(x^*) - \nu x^*) + y^* - y^{k+1}, y^{k+1} - y^*\rangle$$

$$- 2\langle \nu^{-1}(y_g^k + z_g^k) + x^{k+1}, y^{k+1} - y^*\rangle - \frac{1}{\theta}\|y^{k+1} - y^k\|^2$$

$$= \frac{1}{\theta}\|y^k - y^*\|^2 + 2\beta\langle \nabla F(x_g^k) - \nu x_g^k - (\nabla F(x^*) - \nu x^*), y^{k+1} - y^*\rangle - 2\beta\|y^{k+1} - y^*\|^2$$

$$- 2\langle \nu^{-1}(y_g^k + z_g^k) + x^{k+1}, y^{k+1} - y^*\rangle - \frac{1}{\theta}\|y^{k+1} - y^k\|^2$$

$$\leq \frac{1}{\theta}\|y^k - y^*\|^2 + \beta\|\nabla F(x_g^k) - \nu x_g^k - (\nabla F(x^*) - \nu x^*)\|^2 - \beta\|y^{k+1} - y^*\|^2$$

$$- 2\langle \nu^{-1}(y_g^k + z_g^k) + x^{k+1}, y^{k+1} - y^*\rangle - \frac{1}{\theta}\|y^{k+1} - y^k\|^2.$$

Function $F(x) - \frac{\nu}{2}\|x\|^2$ is convex and $L$-smooth, which implies

$$\frac{1}{\theta}\|y^{k+1} - y^*\|^2 \leq \frac{1}{\theta}\|y^k - y^*\|^2 + 2\beta L\left(\mathrm{D}_F(x_g^k, x^*) - \frac{\nu}{2}\|x_g^k - x^*\|^2\right) - \beta\|y^{k+1} - y^*\|^2$$

$$- 2\langle \nu^{-1}(y_g^k + z_g^k) + x^{k+1}, y^{k+1} - y^*\rangle - \frac{1}{\theta}\|y^{k+1} - y^k\|^2.$$

Using $\beta$ definition (38) we get

$$\frac{1}{\theta}\|y^{k+1} - y^*\|^2 \leq \frac{1}{\theta}\|y^k - y^*\|^2 + \mathrm{D}_F(x_g^k, x^*) - \frac{\nu}{2}\|x_g^k - x^*\|^2 - \beta\|y^{k+1} - y^*\|^2$$

$$- 2\langle \nu^{-1}(y_g^k + z_g^k) + x^{k+1}, y^{k+1} - y^*\rangle - \frac{1}{\theta}\|y^{k+1} - y^k\|^2.$$

Using optimality condition (13) we get

$$\frac{1}{\theta}\|y^{k+1} - y^*\|^2 \leq \frac{1}{\theta}\|y^k - y^*\|^2 + \mathrm{D}_F(x_g^k, x^*) - \frac{\nu}{2}\|x_g^k - x^*\|^2 - \beta\|y^{k+1} - y^*\|^2$$

$$- 2\nu^{-1}\langle y_g^k + z_g^k - (y^* + z^*), y^{k+1} - y^*\rangle - 2\langle x^{k+1} - x^*, y^{k+1} - y^*\rangle - \frac{1}{\theta}\|y^{k+1} - y^k\|^2.$$

Using (37) together with $\sigma_1$ definition (39) we get

$$
\begin{aligned}
\frac{1}{\theta}\|y^{k+1} - y^*\|^2 &\leq \frac{1}{\theta}\|y^k - y^*\|^2 + \mathrm{D}_F(x_g^k, x^*) - \frac{\nu}{2}\|x_g^k - x^*\|^2 - \frac{\beta}{2}\|y^{k+1} - y^*\|^2 \\
&\quad + \frac{\beta(1 - \sigma_2/2)}{2\sigma_2}\|y_f^k - y^*\|^2 - \frac{\beta}{2\sigma_2}\|y_f^{k+1} - y^*\|^2 - \frac{\beta}{4}\|y_g^k - y^*\|^2 + \frac{\beta(\sigma_2 - \sigma_1)}{2}\|y^{k+1} - y^k\|^2 \\
&\quad - 2\nu^{-1}\langle y_g^k + z_g^k - (y^* + z^*), y^{k+1} - y^*\rangle - 2\langle x^{k+1} - x^*, y^{k+1} - y^*\rangle - \frac{1}{\theta}\|y^{k+1} - y^k\|^2 \\
&\leq \frac{1}{\theta}\|y^k - y^*\|^2 - \frac{\beta}{2}\|y^{k+1} - y^*\|^2 + \frac{\beta(1 - \sigma_2/2)}{2\sigma_2}\|y_f^k - y^*\|^2 - \frac{\beta}{2\sigma_2}\|y_f^{k+1} - y^*\|^2 \\
&\quad + \mathrm{D}_F(x_g^k, x^*) - \frac{\nu}{2}\|x_g^k - x^*\|^2 - \frac{\beta}{4}\|y_g^k - y^*\|^2 + \left(\frac{\beta\sigma_2^2}{4} - \frac{1}{\theta}\right)\|y^{k+1} - y^k\|^2 \\
&\quad - 2\nu^{-1}\langle y_g^k + z_g^k - (y^* + z^*), y^{k+1} - y^*\rangle - 2\langle x^{k+1} - x^*, y^{k+1} - y^*\rangle.
\end{aligned}
$$

Rearranging gives

$$
\begin{aligned}
\left(\frac{1}{\theta} + \frac{\beta}{2}\right)&\|y^{k+1} - y^*\|^2 + \frac{\beta}{2\sigma_2}\|y_f^{k+1} - y^*\|^2 \\
&\leq \frac{1}{\theta}\|y^k - y^*\|^2 + \frac{\beta(1 - \sigma_2/2)}{2\sigma_2}\|y_f^k - y^*\|^2 + \mathrm{D}_F(x_g^k, x^*) - \frac{\nu}{2}\|x_g^k - x^*\|^2 - 2\langle x^{k+1} - x^*, y^{k+1} - y^*\rangle \\
&\quad - 2\nu^{-1}\langle y_g^k + z_g^k - (y^* + z^*), y^{k+1} - y^*\rangle - \frac{\beta}{4}\|y_g^k - y^*\|^2 + \left(\frac{\beta\sigma_2^2}{4} - \frac{1}{\theta}\right)\|y^{k+1} - y^k\|^2.
\end{aligned}
$$

$\square$

**Lemma 6.** *The following inequality holds:*

$$\|m^k\|_{\mathbf{P}}^2 \le 8\chi^2\gamma^2\nu^{-2}\|y_g^k + z_g^k\|_{\mathbf{P}}^2 + 4\chi(1 - (4\chi)^{-1})\|m^k\|_{\mathbf{P}}^2 - 4\chi\|m^{k+1}\|_{\mathbf{P}}^2. \tag{41}$$

*Proof.* Using Line 12 of Algorithm 1 we get

$$\begin{aligned}
\|m^{k+1}\|_{\mathbf{P}}^2 &= \|\gamma\nu^{-1}(y_g^k + z_g^k) + m^k - (\mathbf{W}(k) \otimes \mathbf{I}_d)\left[\gamma\nu^{-1}(y_g^k + z_g^k) + m^k\right]\|_{\mathbf{P}}^2 \\
&= \|\mathbf{P}\left[\gamma\nu^{-1}(y_g^k + z_g^k) + m^k\right] - (\mathbf{W}(k) \otimes \mathbf{I}_d)\mathbf{P}\left[\gamma\nu^{-1}(y_g^k + z_g^k) + m^k\right]\|^2.
\end{aligned}$$

Using property (3) we obtain

$$\|m^{k+1}\|_{\mathbf{P}}^2 \le (1 - \chi^{-1})\|m^k + \gamma\nu^{-1}(y_g^k + z_g^k)\|_{\mathbf{P}}^2.$$

Using inequality $\|a + b\|^2 \le (1 + c)\|a\|^2 + (1 + c^{-1})\|b\|^2$ with $c = \frac{1}{2(\chi-1)}$ we get

$$\begin{aligned}
\|m^{k+1}\|_{\mathbf{P}}^2 &\le (1 - \chi^{-1})\left[\left(1 + \frac{1}{2(\chi-1)}\right)\|m^k\|_{\mathbf{P}}^2 + (1 + 2(\chi-1))\gamma^2\nu^{-2}\|y_g^k + z_g^k\|_{\mathbf{P}}^2\right] \\
&\le (1 - (2\chi)^{-1})\|m^k\|_{\mathbf{P}}^2 + 2\chi\gamma^2\nu^{-2}\|y_g^k + z_g^k\|_{\mathbf{P}}^2.
\end{aligned}$$

Rearranging gives

$$\|m^k\|_{\mathbf{P}}^2 \le 8\chi^2\gamma^2\nu^{-2}\|y_g^k + z_g^k\|_{\mathbf{P}}^2 + 4\chi(1 - (4\chi)^{-1})\|m^k\|_{\mathbf{P}}^2 - 4\chi\|m^{k+1}\|_{\mathbf{P}}^2.$$

$\square$

**Lemma 7.** *Let $\hat{z}^k$ be defined as follows:*

$$\hat{z}^k = z^k - \mathbf{P}m^k. \tag{42}$$

*Then the following inequality holds:*

$$\begin{aligned}
\frac{1}{\gamma}\|\hat{z}^{k+1} - z^*\|^2 + \frac{4}{3\gamma}\|m^{k+1}\|_{\mathbf{P}}^2 &\le \left(\frac{1}{\gamma} - \delta\right)\|\hat{z}^k - z^*\|^2 + \left(1 - (4\chi)^{-1} + \frac{3\gamma\delta}{2}\right)\frac{4}{3\gamma}\|m^k\|_{\mathbf{P}}^2 \\
&\quad - 2\nu^{-1}\langle y_g^k + z_g^k - (y^* + z^*), z^k - z^*\rangle + \gamma\nu^{-2}(1 + 6\chi)\|y_g^k + z_g^k\|_{\mathbf{P}}^2 \\
&\quad + 2\delta\|z_g^k - z^*\|^2 + \left(2\gamma\delta^2 - \delta\right)\|z_g^k - z^k\|^2.
\end{aligned} \tag{43}$$

*Proof.*

$$\frac{1}{\gamma}\|\hat{z}^{k+1} - z^*\|^2 = \frac{1}{\gamma}\|\hat{z}^k - z^*\|^2 + \frac{2}{\gamma}\langle \hat{z}^{k+1} - \hat{z}^k, \hat{z}^k - z^*\rangle + \frac{1}{\gamma}\|\hat{z}^{k+1} - \hat{z}^k\|^2.$$

Lines 11 and 12 of Algorithm 1 together with $\hat{z}^k$ definition (42) imply

$$\hat{z}^{k+1} - \hat{z}^k = \gamma\delta(z_g^k - z^k) - \gamma\nu^{-1}\mathbf{P}(y_g^k + z_g^k).$$

Hence,

$$\begin{aligned}
\frac{1}{\gamma}\|\hat{z}^{k+1} - z^*\|^2 &= \frac{1}{\gamma}\|\hat{z}^k - z^*\|^2 + 2\delta\langle z_g^k - z^k, \hat{z}^k - z^*\rangle - 2\nu^{-1}\langle \mathbf{P}(y_g^k + z_g^k), \hat{z}^k - z^*\rangle + \frac{1}{\gamma}\|\hat{z}^{k+1} - \hat{z}^k\|^2 \\
&= \frac{1}{\gamma}\|\hat{z}^k - z^*\|^2 + \delta\|z_g^k - \mathbf{P}m^k - z^*\|^2 - \delta\|\hat{z}^k - z^*\|^2 - \delta\|z_g^k - z^k\|^2 \\
&\quad - 2\nu^{-1}\langle \mathbf{P}(y_g^k + z_g^k), \hat{z}^k - z^*\rangle + \gamma\|\delta(z_g^k - z^k) - \nu^{-1}\mathbf{P}(y_g^k + z_g^k)\|^2 \\
&\le \left(\frac{1}{\gamma} - \delta\right)\|\hat{z}^k - z^*\|^2 + 2\delta\|z_g^k - z^*\|^2 + 2\delta\|m^k\|_{\mathbf{P}}^2 - \delta\|z_g^k - z^k\|^2 \\
&\quad - 2\nu^{-1}\langle \mathbf{P}(y_g^k + z_g^k), \hat{z}^k - z^*\rangle + 2\gamma\delta^2\|z_g^k - z^k\|^2 + \gamma\|\nu^{-1}\mathbf{P}(y_g^k + z_g^k)\|^2 \\
&\le \left(\frac{1}{\gamma} - \delta\right)\|\hat{z}^k - z^*\|^2 + 2\delta\|z_g^k - z^*\|^2 + \left(2\gamma\delta^2 - \delta\right)\|z_g^k - z^k\|^2 \\
&\quad - 2\nu^{-1}\langle \mathbf{P}(y_g^k + z_g^k), z^k - z^*\rangle + \gamma\|\nu^{-1}\mathbf{P}(y_g^k + z_g^k)\|^2 + 2\delta\|m^k\|_{\mathbf{P}}^2 + 2\nu^{-1}\langle \mathbf{P}(y_g^k + z_g^k), m^k\rangle.
\end{aligned}$$

Using the fact that $z^k \in \mathcal{L}^\perp$ for all $k = 0, 1, 2 \ldots$ and optimality condition (14) we get

$$\frac{1}{\gamma}\|\hat{z}^{k+1} - z^*\|^2 \leq \left(\frac{1}{\gamma} - \delta\right)\|\hat{z}^k - z^*\|^2 + 2\delta\|z_g^k - z^*\|^2 + \left(2\gamma\delta^2 - \delta\right)\|z_g^k - z^k\|^2$$
$$- 2\nu^{-1}\langle y_g^k + z_g^k - (y^* + z^*), z^k - z^*\rangle + \gamma\nu^{-2}\|y_g^k + z_g^k\|_{\mathbf{P}}^2$$
$$+ 2\delta\|m^k\|_{\mathbf{P}}^2 + 2\nu^{-1}\langle \mathbf{P}(y_g^k + z_g^k), m^k\rangle.$$

Using Young's inequality we get

$$\frac{1}{\gamma}\|\hat{z}^{k+1} - z^*\|^2 \leq \left(\frac{1}{\gamma} - \delta\right)\|\hat{z}^k - z^*\|^2 + 2\delta\|z_g^k - z^*\|^2 + \left(2\gamma\delta^2 - \delta\right)\|z_g^k - z^k\|^2$$
$$- 2\nu^{-1}\langle y_g^k + z_g^k - (y^* + z^*), z^k - z^*\rangle + \gamma\nu^{-2}\|y_g^k + z_g^k\|_{\mathbf{P}}^2$$
$$+ 2\delta\|m^k\|_{\mathbf{P}}^2 + 3\gamma\chi\nu^{-2}\|y_g^k + z_g^k\|_{\mathbf{P}}^2 + \frac{1}{3\gamma\chi}\|m^k\|_{\mathbf{P}}^2.$$

Using (41) we get

$$\frac{1}{\gamma}\|\hat{z}^{k+1} - z^*\|^2 \leq \left(\frac{1}{\gamma} - \delta\right)\|\hat{z}^k - z^*\|^2 + 2\delta\|z_g^k - z^*\|^2 + \left(2\gamma\delta^2 - \delta\right)\|z_g^k - z^k\|^2$$
$$- 2\nu^{-1}\langle y_g^k + z_g^k - (y^* + z^*), z^k - z^*\rangle + \gamma\nu^{-2}\|y_g^k + z_g^k\|_{\mathbf{P}}^2$$
$$+ 2\delta\|m^k\|_{\mathbf{P}}^2 + 6\gamma\nu^{-2}\chi\|y_g^k + z_g^k\|_{\mathbf{P}}^2 + \frac{4(1 - (4\chi)^{-1})}{3\gamma}\|m^k\|_{\mathbf{P}}^2 - \frac{4}{3\gamma}\|m^{k+1}\|_{\mathbf{P}}^2$$
$$= \left(\frac{1}{\gamma} - \delta\right)\|\hat{z}^k - z^*\|^2 + 2\delta\|z_g^k - z^*\|^2 + \left(2\gamma\delta^2 - \delta\right)\|z_g^k - z^k\|^2$$
$$- 2\nu^{-1}\langle y_g^k + z_g^k - (y^* + z^*), z^k - z^*\rangle + \gamma\nu^{-2}\left(1 + 6\chi\right)\|y_g^k + z_g^k\|_{\mathbf{P}}^2$$
$$+ \left(1 - (4\chi)^{-1} + \frac{3\gamma\delta}{2}\right)\frac{4}{3\gamma}\|m^k\|_{\mathbf{P}}^2 - \frac{4}{3\gamma}\|m^{k+1}\|_{\mathbf{P}}^2.$$

$\square$

**Lemma 8.** *The following inequality holds:*
$$2\langle y_g^k + z_g^k - (y^* + z^*), y^k + z^k - (y^* + z^*)\rangle$$
$$\geq 2\|y_g^k + z_g^k - (y^* + z^*)\|^2 + \frac{(1 - \sigma_2/2)}{\sigma_2}\left(\|y_g^k + z_g^k - (y^* + z^*)\|^2 - \|y_f^k + z_f^k - (y^* + z^*)\|^2\right).$$
(44)

*Proof.*
$$2\langle y_g^k + z_g^k - (y^* + z^*), y^k + z^k - (y^* + z^*)\rangle$$
$$= 2\|y_g^k + z_g^k - (y^* + z^*)\|^2 + 2\langle y_g^k + z_g^k - (y^* + z^*), y^k + z^k - (y_g^k + z_g^k)\rangle.$$

Using Lines 7 and 10 of Algorithm 1 we get
$$2\langle y_g^k + z_g^k - (y^* + z^*), y^k + z^k - (y^* + z^*)\rangle$$
$$= 2\|y_g^k + z_g^k - (y^* + z^*)\|^2 + \frac{2(1 - \sigma_1)}{\sigma_1}\langle y_g^k + z_g^k - (y^* + z^*), y_g^k + z_g^k - (y_f^k + z_f^k)\rangle$$
$$= 2\|y_g^k + z_g^k - (y^* + z^*)\|^2$$
$$+ \frac{(1 - \sigma_1)}{\sigma_1}\left(\|y_g^k + z_g^k - (y^* + z^*)\|^2 + \|y_g^k + z_g^k - (y_f^k + z_f^k)\|^2 - \|y_f^k + z_f^k - (y^* + z^*)\|^2\right)$$
$$\geq 2\|y_g^k + z_g^k - (y^* + z^*)\|^2 + \frac{(1 - \sigma_1)}{\sigma_1}\left(\|y_g^k + z_g^k - (y^* + z^*)\|^2 - \|y_f^k + z_f^k - (y^* + z^*)\|^2\right).$$

Using $\sigma_1$ definition (39) we get
$$2\langle y_g^k + z_g^k - (y^* + z^*), y^k + z^k - (y^* + z^*)\rangle$$
$$\geq 2\|y_g^k + z_g^k - (y^* + z^*)\|^2 + \frac{(1 - \sigma_2/2)}{\sigma_2}\left(\|y_g^k + z_g^k - (y^* + z^*)\|^2 - \|y_f^k + z_f^k - (y^* + z^*)\|^2\right).$$

$\square$

**Lemma 9.** *Let $\zeta$ be defined by*

$$\zeta = 1/2. \tag{45}$$

*Then the following inequality holds:*

$$-2\langle y^{k+1} - y^k, y_g^k + z_g^k - (y^* + z^*)\rangle$$

$$\leq \frac{1}{\sigma_2}\|y_g^k + z_g^k - (y^* + z^*)\|^2 - \frac{1}{\sigma_2}\|y_f^{k+1} + z_f^{k+1} - (y^* + z^*)\|^2 \tag{46}$$

$$+ 2\sigma_2\|y^{k+1} - y^k\|^2 - \frac{1}{2\sigma_2\chi}\|y_g^k + z_g^k\|_{\mathbf{P}}^2.$$

*Proof.*

$$\|y_f^{k+1} + z_f^{k+1} - (y^* + z^*)\|^2$$

$$= \|y_g^k + z_g^k - (y^* + z^*)\|^2 + 2\langle y_f^{k+1} + z_f^{k+1} - (y_g^k + z_g^k), y_g^k + z_g^k - (y^* + z^*)\rangle$$

$$+ \|y_f^{k+1} + z_f^{k+1} - (y_g^k + z_g^k)\|^2$$

$$\leq \|y_g^k + z_g^k - (y^* + z^*)\|^2 + 2\langle y_f^{k+1} + z_f^{k+1} - (y_g^k + z_g^k), y_g^k + z_g^k - (y^* + z^*)\rangle$$

$$+ 2\|y_f^{k+1} - y_g^k\|^2 + 2\|z_f^{k+1} - z_g^k\|^2.$$

Using Line 9 of Algorithm 1 we get

$$\|y_f^{k+1} + z_f^{k+1} - (y^* + z^*)\|^2$$

$$\leq \|y_g^k + z_g^k - (y^* + z^*)\|^2 + 2\sigma_2\langle y^{k+1} - y^k, y_g^k + z_g^k - (y^* + z^*)\rangle + 2\sigma_2^2\|y^{k+1} - y^k\|^2$$

$$+ 2\langle z_f^{k+1} - z_g^k, y_g^k + z_g^k - (y^* + z^*)\rangle + 2\|z_f^{k+1} - z_g^k\|^2.$$

Using Line 13 of Algorithm 1 and optimality condition (14) we get

$$\|y_f^{k+1} + z_f^{k+1} - (y^* + z^*)\|^2$$

$$\leq \|y_g^k + z_g^k - (y^* + z^*)\|^2 + 2\sigma_2\langle y^{k+1} - y^k, y_g^k + z_g^k - (y^* + z^*)\rangle + 2\sigma_2^2\|y^{k+1} - y^k\|^2$$

$$- 2\zeta\langle (\mathbf{W}(k) \otimes \mathbf{I}_d)(y_g^k + z_g^k), y_g^k + z_g^k - (y^* + z^*)\rangle + 2\zeta^2\|(\mathbf{W}(k) \otimes \mathbf{I}_d)(y_g^k + z_g^k)\|^2$$

$$= \|y_g^k + z_g^k - (y^* + z^*)\|^2 + 2\sigma_2\langle y^{k+1} - y^k, y_g^k + z_g^k - (y^* + z^*)\rangle + 2\sigma_2^2\|y^{k+1} - y^k\|^2$$

$$- 2\zeta\langle (\mathbf{W}(k) \otimes \mathbf{I}_d)(y_g^k + z_g^k), y_g^k + z_g^k\rangle + 2\zeta^2\|(\mathbf{W}(k) \otimes \mathbf{I}_d)(y_g^k + z_g^k)\|^2.$$

Using $\zeta$ definition (45) we get

$$\|y_f^{k+1} + z_f^{k+1} - (y^* + z^*)\|^2$$

$$\leq \|y_g^k + z_g^k - (y^* + z^*)\|^2 + 2\sigma_2\langle y^{k+1} - y^k, y_g^k + z_g^k - (y^* + z^*)\rangle + 2\sigma_2^2\|y^{k+1} - y^k\|^2$$

$$- \langle (\mathbf{W}(k) \otimes \mathbf{I}_d)(y_g^k + z_g^k), y_g^k + z_g^k\rangle + \frac{1}{2}\|(\mathbf{W}(k) \otimes \mathbf{I}_d)(y_g^k + z_g^k)\|^2$$

$$= \|y_g^k + z_g^k - (y^* + z^*)\|^2 + 2\sigma_2\langle y^{k+1} - y^k, y_g^k + z_g^k - (y^* + z^*)\rangle + 2\sigma_2^2\|y^{k+1} - y^k\|^2$$

$$- \frac{1}{2}\|(\mathbf{W}(k) \otimes \mathbf{I}_d)(y_g^k + z_g^k)\|^2 - \frac{1}{2}\|y_g^k + z_g^k\|^2 + \frac{1}{2}\|(\mathbf{W}(k) \otimes \mathbf{I}_d)(y_g^k + z_g^k) - (y_g^k + z_g^k)\|^2$$

$$+ \frac{1}{2}\|(\mathbf{W}(k) \otimes \mathbf{I}_d)(y_g^k + z_g^k)\|^2$$

$$\leq \|y_g^k + z_g^k - (y^* + z^*)\|^2 + 2\sigma_2\langle y^{k+1} - y^k, y_g^k + z_g^k - (y^* + z^*)\rangle + 2\sigma_2^2\|y^{k+1} - y^k\|^2$$

$$- \frac{1}{2}\|y_g^k + z_g^k\|_{\mathbf{P}}^2 + \frac{1}{2}\|(\mathbf{W}(k) \otimes \mathbf{I}_d)(y_g^k + z_g^k) - (y_g^k + z_g^k)\|_{\mathbf{P}}^2.$$

$$= \|y_g^k + z_g^k - (y^* + z^*)\|^2 + 2\sigma_2\langle y^{k+1} - y^k, y_g^k + z_g^k - (y^* + z^*)\rangle + 2\sigma_2^2\|y^{k+1} - y^k\|^2$$

$$- \frac{1}{2}\|y_g^k + z_g^k\|_{\mathbf{P}}^2 + \frac{1}{2}\|(\mathbf{W}(k) \otimes \mathbf{I}_d)\mathbf{P}(y_g^k + z_g^k) - \mathbf{P}(y_g^k + z_g^k)\|^2.$$

Using condition (3) we get

$$\|y_f^{k+1} + z_f^{k+1} - (y^* + z^*)\|^2$$

$$\leq \|y_g^k + z_g^k - (y^* + z^*)\|^2 + 2\sigma_2 \langle y^{k+1} - y^k, y_g^k + z_g^k - (y^* + z^*)\rangle + 2\sigma_2^2 \|y^{k+1} - y^k\|^2$$
$$- (2\chi)^{-1}\|y_g^k + z_g^k\|_{\mathbf{P}}^2.$$

Rearranging gives

$$-2\langle y^{k+1} - y^k, y_g^k + z_g^k - (y^* + z^*)\rangle$$
$$\leq \frac{1}{\sigma_2}\|y_g^k + z_g^k - (y^* + z^*)\|^2 - \frac{1}{\sigma_2}\|y_f^{k+1} + z_f^{k+1} - (y^* + z^*)\|^2$$
$$+ 2\sigma_2\|y^{k+1} - y^k\|^2 - \frac{1}{2\sigma_2\chi}\|y_g^k + z_g^k\|_{\mathbf{P}}^2.$$

$\square$

**Lemma 10.** *Let $\delta$ be defined as follows:*

$$\delta = \frac{1}{17L}. \tag{47}$$

*Let $\gamma$ be defined as follows:*

$$\gamma = \frac{\nu}{14\sigma_2\chi^2}. \tag{48}$$

*Let $\theta$ be defined as follows:*

$$\theta = \frac{\nu}{4\sigma_2}. \tag{49}$$

*Let $\sigma_2$ be defined as follows:*

$$\sigma_2 = \frac{\sqrt{\mu}}{16\chi\sqrt{L}}. \tag{50}$$

*Let $\Psi_{yz}^k$ be the following Lyapunov function*

$$\Psi_{yz}^k = \left(\frac{1}{\theta} + \frac{\beta}{2}\right)\|y^k - y^*\|^2 + \frac{\beta}{2\sigma_2}\|y_f^k - y^*\|^2 + \frac{1}{\gamma}\|\hat{z}^k - z^*\|^2$$
$$+ \frac{4}{3\gamma}\|m^k\|_{\mathbf{P}}^2 + \frac{\nu^{-1}}{\sigma_2}\|y_f^k + z_f^k - (y^* + z^*)\|^2. \tag{51}$$

*Then the following inequality holds:*

$$\Psi_{yz}^{k+1} \leq \left(1 - \frac{\sqrt{\mu}}{32\chi\sqrt{L}}\right)\Psi_{yz}^k + \mathrm{D}_F(x_g^k, x^*) - \frac{\nu}{2}\|x_g^k - x^*\|^2 - 2\langle x^{k+1} - x^*, y^{k+1} - y^*\rangle. \tag{52}$$

*Proof.* Combining (40) and (43) gives

$$\left(\frac{1}{\theta} + \frac{\beta}{2}\right)\|y^{k+1} - y^*\|^2 + \frac{\beta}{2\sigma_2}\|y_f^{k+1} - y^*\|^2 + \frac{1}{\gamma}\|\hat{z}^{k+1} - z^*\|^2 + \frac{4}{3\gamma}\|m^{k+1}\|_{\mathbf{P}}^2$$
$$\leq \left(\frac{1}{\gamma} - \delta\right)\|\hat{z}^k - z^*\|^2 + \left(1 - (4\chi)^{-1} + \frac{3\gamma\delta}{2}\right)\frac{4}{3\gamma}\|m^k\|_{\mathbf{P}}^2 + \frac{1}{\theta}\|y^k - y^*\|^2 + \frac{\beta(1 - \sigma_2/2)}{2\sigma_2}\|y_f^k - y^*\|^2$$
$$- 2\nu^{-1}\langle y_g^k + z_g^k - (y^* + z^*), y^k + z^k - (y^* + z^*)\rangle - 2\nu^{-1}\langle y_g^k + z_g^k - (y^* + z^*), y^{k+1} - y^k\rangle$$
$$+ \gamma\nu^{-2}(1 + 6\chi)\|y_g^k + z_g^k\|_{\mathbf{P}}^2 + \left(\frac{\beta\sigma_2^2}{4} - \frac{1}{\theta}\right)\|y^{k+1} - y^k\|^2 + 2\delta\|z_g^k - z^*\|^2 - \frac{\beta}{4}\|y_g^k - y^*\|^2$$
$$+ \mathrm{D}_F(x_g^k, x^*) - \frac{\nu}{2}\|x_g^k - x^*\|^2 - 2\langle x^{k+1} - x^*, y^{k+1} - y^*\rangle + (2\gamma\delta^2 - \delta)\|z_g^k - z^k\|^2.$$

Using (44) and (46) we get

$$\left(\frac{1}{\theta} + \frac{\beta}{2}\right)\|y^{k+1} - y^*\|^2 + \frac{\beta}{2\sigma_2}\|y_f^{k+1} - y^*\|^2 + \frac{1}{\gamma}\|\hat{z}^{k+1} - z^*\|^2 + \frac{4}{3\gamma}\|m^{k+1}\|_{\mathbf{P}}^2$$
$$\leq \left(\frac{1}{\gamma} - \delta\right)\|\hat{z}^k - z^*\|^2 + \left(1 - (4\chi)^{-1} + \frac{3\gamma\delta}{2}\right)\frac{4}{3\gamma}\|m^k\|_{\mathbf{P}}^2 + \frac{1}{\theta}\|y^k - y^*\|^2 + \frac{\beta(1 - \sigma_2/2)}{2\sigma_2}\|y_f^k - y^*\|^2$$

$$-2\nu^{-1}\|y_g^k + z_g^k - (y^* + z^*)\|^2 + \frac{\nu^{-1}(1 - \sigma_2/2)}{\sigma_2}\left(\|y_f^k + z_f^k - (y^* + z^*)\|^2 - \|y_g^k + z_g^k - (y^* + z^*)\|^2\right)$$

$$+\frac{\nu^{-1}}{\sigma_2}\|y_g^k + z_g^k - (y^* + z^*)\|^2 - \frac{\nu^{-1}}{\sigma_2}\|y_f^{k+1} + z_f^{k+1} - (y^* + z^*)\|^2 + 2\nu^{-1}\sigma_2\|y^{k+1} - y^k\|^2$$

$$-\frac{\nu^{-1}}{2\sigma_2\chi}\|y_g^k + z_g^k\|_{\mathbf{P}}^2 + \gamma\nu^{-2}(1 + 6\chi)\|y_g^k + z_g^k\|_{\mathbf{P}}^2 + \left(\frac{\beta\sigma_2^2}{4} - \frac{1}{\theta}\right)\|y^{k+1} - y^k\|^2 + 2\delta\|z_g^k - z^*\|^2$$

$$-\frac{\beta}{4}\|y_g^k - y^*\|^2 + \mathrm{D}_F(x_g^k, x^*) - \frac{\nu}{2}\|x_g^k - x^*\|^2 - 2\langle x^{k+1} - x^*, y^{k+1} - y^*\rangle + (2\gamma\delta^2 - \delta)\|z_g^k - z^k\|^2$$

$$= \left(\frac{1}{\gamma} - \delta\right)\|\hat{z}^k - z^*\|^2 + \left(1 - (4\chi)^{-1} + \frac{3\gamma\delta}{2}\right)\frac{4}{3\gamma}\|m^k\|_{\mathbf{P}}^2 + \frac{1}{\theta}\|y^k - y^*\|^2 + \frac{\beta(1 - \sigma_2/2)}{2\sigma_2}\|y_f^k - y^*\|^2$$

$$+\frac{\nu^{-1}(1 - \sigma_2/2)}{\sigma_2}\|y_f^k + z_f^k - (y^* + z^*)\|^2 - \frac{\nu^{-1}}{\sigma_2}\|y_f^{k+1} + z_f^{k+1} - (y^* + z^*)\|^2$$

$$+ 2\delta\|z_g^k - z^*\|^2 - \frac{\beta}{4}\|y_g^k - y^*\|^2 + \nu^{-1}\left(\frac{1}{\sigma_2} - \frac{(1 - \sigma_2/2)}{\sigma_2} - 2\right)\|y_g^k + z_g^k - (y^* + z^*)\|^2$$

$$+ \left(\gamma\nu^{-2}(1 + 6\chi) - \frac{\nu^{-1}}{2\sigma_2\chi}\right)\|y_g^k + z_g^k\|_{\mathbf{P}}^2 + \left(\frac{\beta\sigma_2^2}{4} + 2\nu^{-1}\sigma_2 - \frac{1}{\theta}\right)\|y^{k+1} - y^k\|^2$$

$$+ (2\gamma\delta^2 - \delta)\|z_g^k - z^k\|^2 + \mathrm{D}_F(x_g^k, x^*) - \frac{\nu}{2}\|x_g^k - x^*\|^2 - 2\langle x^{k+1} - x^*, y^{k+1} - y^*\rangle$$

$$= \left(\frac{1}{\gamma} - \delta\right)\|\hat{z}^k - z^*\|^2 + \left(1 - (4\chi)^{-1} + \frac{3\gamma\delta}{2}\right)\frac{4}{3\gamma}\|m^k\|_{\mathbf{P}}^2 + \frac{1}{\theta}\|y^k - y^*\|^2 + \frac{\beta(1 - \sigma_2/2)}{2\sigma_2}\|y_f^k - y^*\|^2$$

$$+\frac{\nu^{-1}(1 - \sigma_2/2)}{\sigma_2}\|y_f^k + z_f^k - (y^* + z^*)\|^2 - \frac{\nu^{-1}}{\sigma_2}\|y_f^{k+1} + z_f^{k+1} - (y^* + z^*)\|^2$$

$$+ 2\delta\|z_g^k - z^*\|^2 - \frac{\beta}{4}\|y_g^k - y^*\|^2 - \frac{3\nu^{-1}}{2}\|y_g^k + z_g^k - (y^* + z^*)\|^2 + (2\gamma\delta^2 - \delta)\|z_g^k - z^k\|^2$$

$$+ \left(\gamma\nu^{-2}(1 + 6\chi) - \frac{\nu^{-1}}{2\sigma_2\chi}\right)\|y_g^k + z_g^k\|_{\mathbf{P}}^2 + \left(\frac{\beta\sigma_2^2}{4} + 2\nu^{-1}\sigma_2 - \frac{1}{\theta}\right)\|y^{k+1} - y^k\|^2$$

$$+ \mathrm{D}_F(x_g^k, x^*) - \frac{\nu}{2}\|x_g^k - x^*\|^2 - 2\langle x^{k+1} - x^*, y^{k+1} - y^*\rangle.$$

Using $\beta$ definition (38) and $\nu$ definition (34) we get

$$\left(\frac{1}{\theta} + \frac{\beta}{2}\right)\|y^{k+1} - y^*\|^2 + \frac{\beta}{2\sigma_2}\|y_f^{k+1} - y^*\|^2 + \frac{1}{\gamma}\|\hat{z}^{k+1} - z^*\|^2 + \frac{4}{3\gamma}\|m^{k+1}\|_{\mathbf{P}}^2$$

$$\leq \left(\frac{1}{\gamma} - \delta\right)\|\hat{z}^k - z^*\|^2 + \left(1 - (4\chi)^{-1} + \frac{3\gamma\delta}{2}\right)\frac{4}{3\gamma}\|m^k\|_{\mathbf{P}}^2 + \frac{1}{\theta}\|y^k - y^*\|^2 + \frac{\beta(1 - \sigma_2/2)}{2\sigma_2}\|y_f^k - y^*\|^2$$

$$+\frac{\nu^{-1}(1 - \sigma_2/2)}{\sigma_2}\|y_f^k + z_f^k - (y^* + z^*)\|^2 - \frac{\nu^{-1}}{\sigma_2}\|y_f^{k+1} + z_f^{k+1} - (y^* + z^*)\|^2$$

$$+ 2\delta\|z_g^k - z^*\|^2 - \frac{1}{8L}\|y_g^k - y^*\|^2 - \frac{3}{\mu}\|y_g^k + z_g^k - (y^* + z^*)\|^2 + (2\gamma\delta^2 - \delta)\|z_g^k - z^k\|^2$$

$$+ \left(\gamma\nu^{-2}(1 + 6\chi) - \frac{\nu^{-1}}{2\sigma_2\chi}\right)\|y_g^k + z_g^k\|_{\mathbf{P}}^2 + \left(\frac{\beta\sigma_2^2}{4} + 2\nu^{-1}\sigma_2 - \frac{1}{\theta}\right)\|y^{k+1} - y^k\|^2$$

$$+ \mathrm{D}_F(x_g^k, x^*) - \frac{\nu}{2}\|x_g^k - x^*\|^2 - 2\langle x^{k+1} - x^*, y^{k+1} - y^*\rangle.$$

Using $\delta$ definition (47) we get

$$\left(\frac{1}{\theta} + \frac{\beta}{2}\right)\|y^{k+1} - y^*\|^2 + \frac{\beta}{2\sigma_2}\|y_f^{k+1} - y^*\|^2 + \frac{1}{\gamma}\|\hat{z}^{k+1} - z^*\|^2 + \frac{4}{3\gamma}\|m^{k+1}\|_{\mathbf{P}}^2$$

$$\leq \left(\frac{1}{\gamma} - \delta\right)\|\hat{z}^k - z^*\|^2 + \left(1 - (4\chi)^{-1} + \frac{3\gamma\delta}{2}\right)\frac{4}{3\gamma}\|m^k\|_{\mathbf{P}}^2 + \frac{1}{\theta}\|y^k - y^*\|^2 + \frac{\beta(1 - \sigma_2/2)}{2\sigma_2}\|y_f^k - y^*\|^2$$

$$+\frac{\nu^{-1}(1 - \sigma_2/2)}{\sigma_2}\|y_f^k + z_f^k - (y^* + z^*)\|^2 - \frac{\nu^{-1}}{\sigma_2}\|y_f^{k+1} + z_f^{k+1} - (y^* + z^*)\|^2$$

$$+ \left( \gamma \nu^{-2} \left( 1 + 6\chi \right) - \frac{\nu^{-1}}{2\sigma_2 \chi} \right) \| y_g^k + z_g^k \|_{\mathbf{P}}^2 + \left( \frac{\beta \sigma_2^2}{4} + 2\nu^{-1}\sigma_2 - \frac{1}{\theta} \right) \| y^{k+1} - y^k \|^2$$

$$+ \left( 2\gamma\delta^2 - \delta \right) \| z_g^k - z^k \|^2 + \mathrm{D}_F(x_g^k, x^*) - \frac{\nu}{2} \| x_g^k - x^* \|^2 - 2 \langle x^{k+1} - x^*, y^{k+1} - y^* \rangle.$$

Using $\gamma$ definition (48) we get

$$\left( \frac{1}{\theta} + \frac{\beta}{2} \right) \| y^{k+1} - y^* \|^2 + \frac{\beta}{2\sigma_2} \| y_f^{k+1} - y^* \|^2 + \frac{1}{\gamma} \| \hat{z}^{k+1} - z^* \|^2 + \frac{4}{3\gamma} \| m^{k+1} \|_{\mathbf{P}}^2$$

$$\leq \left( \frac{1}{\gamma} - \delta \right) \| \hat{z}^k - z^* \|^2 + \left( 1 - (4\chi)^{-1} + \frac{3\gamma\delta}{2} \right) \frac{4}{3\gamma} \| m^k \|_{\mathbf{P}}^2 + \frac{1}{\theta} \| y^k - y^* \|^2 + \frac{\beta(1 - \sigma_2/2)}{2\sigma_2} \| y_f^k - y^* \|^2$$

$$+ \frac{\nu^{-1}(1 - \sigma_2/2)}{\sigma_2} \| y_f^k + z_f^k - (y^* + z^*) \|^2 - \frac{\nu^{-1}}{\sigma_2} \| y_f^{k+1} + z_f^{k+1} - (y^* + z^*) \|^2$$

$$+ \left( \frac{\beta \sigma_2^2}{4} + 2\nu^{-1}\sigma_2 - \frac{1}{\theta} \right) \| y^{k+1} - y^k \|^2 + \left( 2\gamma\delta^2 - \delta \right) \| z_g^k - z^k \|^2$$

$$+ \mathrm{D}_F(x_g^k, x^*) - \frac{\nu}{2} \| x_g^k - x^* \|^2 - 2 \langle x^{k+1} - x^*, y^{k+1} - y^* \rangle.$$

Using $\theta$ definition together with (34), (38) and (50) gives

$$\left( \frac{1}{\theta} + \frac{\beta}{2} \right) \| y^{k+1} - y^* \|^2 + \frac{\beta}{2\sigma_2} \| y_f^{k+1} - y^* \|^2 + \frac{1}{\gamma} \| \hat{z}^{k+1} - z^* \|^2 + \frac{4}{3\gamma} \| m^{k+1} \|_{\mathbf{P}}^2$$

$$\leq \left( \frac{1}{\gamma} - \delta \right) \| \hat{z}^k - z^* \|^2 + \left( 1 - (4\chi)^{-1} + \frac{3\gamma\delta}{2} \right) \frac{4}{3\gamma} \| m^k \|_{\mathbf{P}}^2 + \frac{1}{\theta} \| y^k - y^* \|^2 + \frac{\beta(1 - \sigma_2/2)}{2\sigma_2} \| y_f^k - y^* \|^2$$

$$+ \frac{\nu^{-1}(1 - \sigma_2/2)}{\sigma_2} \| y_f^k + z_f^k - (y^* + z^*) \|^2 - \frac{\nu^{-1}}{\sigma_2} \| y_f^{k+1} + z_f^{k+1} - (y^* + z^*) \|^2$$

$$+ \left( 2\gamma\delta^2 - \delta \right) \| z_g^k - z^k \|^2 + \mathrm{D}_F(x_g^k, x^*) - \frac{\nu}{2} \| x_g^k - x^* \|^2 - 2 \langle x^{k+1} - x^*, y^{k+1} - y^* \rangle.$$

Using $\gamma$ definition (48) and $\delta$ definition (47) we get

$$\left( \frac{1}{\theta} + \frac{\beta}{2} \right) \| y^{k+1} - y^* \|^2 + \frac{\beta}{2\sigma_2} \| y_f^{k+1} - y^* \|^2 + \frac{1}{\gamma} \| \hat{z}^{k+1} - z^* \|^2 + \frac{4}{3\gamma} \| m^{k+1} \|_{\mathbf{P}}^2$$

$$\leq \left( \frac{1}{\gamma} - \delta \right) \| \hat{z}^k - z^* \|^2 + \left( 1 - (8\chi)^{-1} \right) \frac{4}{3\gamma} \| m^k \|_{\mathbf{P}}^2 + \frac{1}{\theta} \| y^k - y^* \|^2 + \frac{\beta(1 - \sigma_2/2)}{2\sigma_2} \| y_f^k - y^* \|^2$$

$$+ \frac{\nu^{-1}(1 - \sigma_2/2)}{\sigma_2} \| y_f^k + z_f^k - (y^* + z^*) \|^2 - \frac{\nu^{-1}}{\sigma_2} \| y_f^{k+1} + z_f^{k+1} - (y^* + z^*) \|^2$$

$$+ \mathrm{D}_F(x_g^k, x^*) - \frac{\nu}{2} \| x_g^k - x^* \|^2 - 2 \langle x^{k+1} - x^*, y^{k+1} - y^* \rangle.$$

After rearranging and using $\Psi_{yz}^k$ definition (51) we get

$$\Psi_{yz}^{k+1} \leq \max \left\{ (1 + \theta\beta/2)^{-1}, (1 - \gamma\delta), (1 - \sigma_2/2), (1 - (8\chi)^{-1}) \right\} \Psi_{yz}^k$$

$$+ \mathrm{D}_F(x_g^k, x^*) - \frac{\nu}{2} \| x_g^k - x^* \|^2 - 2 \langle x^{k+1} - x^*, y^{k+1} - y^* \rangle$$

$$\leq \left( 1 - \frac{\sqrt{\mu}}{32\chi\sqrt{L}} \right) \Psi_{yz}^k + \mathrm{D}_F(x_g^k, x^*) - \frac{\nu}{2} \| x_g^k - x^* \|^2 - 2 \langle x^{k+1} - x^*, y^{k+1} - y^* \rangle.$$

$\square$

*Proof of Theorem 4.* Combining (36) and (52) gives

$$\Psi_x^{k+1} + \Psi_{yz}^{k+1} \leq \left(1 - \frac{\sqrt{\mu}}{\sqrt{\mu} + 2\sqrt{L}}\right)\Psi_x^k + \left(1 - \frac{\lambda_{\min}\sqrt{\mu}}{32\lambda_{\max}\sqrt{L}}\right)\Psi_{yz}^k$$

$$\leq \left(1 - \frac{\lambda_{\min}\sqrt{\mu}}{32\lambda_{\max}\sqrt{L}}\right)(\Psi_x^k + \Psi_{yz}^k).$$

This implies

$$\Psi_x^k + \Psi_{yz}^k \leq \left(1 - \frac{\lambda_{\min}\sqrt{\mu}}{32\lambda_{\max}\sqrt{L}}\right)^k (\Psi_x^0 + \Psi_{yz}^0).$$

Using $\Psi_x^k$ definition (35) we get

$$\|x^k - x^*\|^2 \leq \eta\Psi_x^k \leq \eta(\Psi_x^k + \Psi_{yz}^k) \leq \left(1 - \frac{\lambda_{\min}\sqrt{\mu}}{32\lambda_{\max}\sqrt{L}}\right)^k \eta(\Psi_x^0 + \Psi_{yz}^0).$$

Choosing $C = \eta(\Psi_x^0 + \Psi_{yz}^0)$ and using the number of iterations

$$k = 32\chi\sqrt{L/\mu}\log\frac{C}{\varepsilon} = \mathcal{O}\left(\chi\sqrt{L/\mu}\log\frac{1}{\epsilon}\right).$$

we get

$$\|x^k - x^*\|^2 \leq \epsilon,$$

which concludes the proof. □