# OpenReview forum: "Lower Bounds and Optimal Algorithms for Smooth and Strongly Convex Decentralized Optimization Over Time-Varying Networks"
_NeurIPS.cc/2021/Conference — NeurIPS 2021 Poster_

### Official Review · Reviewer_qUuQ · 2021-07-06

**Rating:** 5
**Confidence:** 1

**Summary:**

This paper studies a decentralized optimization problem. There is a set of nodes that collectively want to find the minimum of an objective function that is the sum of local costs. The nodes can communicate on underlying communication graph.  The authors study the trade-off communication v.s. precision. They show a lower bound on the quality of a solution that can be attained given a certain communication budget. They then provide two algorithms that attain this lower bound.


**Ethical Concerns:**

NA.

**Limitations And Societal Impact:**

To be more accessible to a non-expert like me, the assumption behind the communication model and the contribution with respect to related work could be better explained.

**Main Review:**

I must say that I have very limited knowledge on this topic so my review has to be taken with a grain of salt. Yet, I find the paper not easy to read in the many technical part. For instance, I am not sure to understand exactly what is the matrix $W(q)$ defined in section 2.4 and what are the implication of Equation (3) -> in section 5.1, it is written "as discussed in Section 2.4, a decentralized communication round can be represented by a multiplication by [...] W(q)". I would have appreciated more details there. On the other hand, I believe that some part can be shorten (what is the interest of Figure 1? Would not it be better to merge Theorem 2 and 3 into one?)

The lower bound (Theorem 1) seem to be the main contribution of the paper. The authors could comment on the impact of choosing $d=+\infty$ here?  Would it change anything?

The algorithmic part is harder to understand for me. For instance, is it obvious that the multi-consensus algorithm falls into the category of algorithms used in the lower bound of Theorem 1? Also, the contribution with respect to ADOM (Kovalev et al 2021) is unclear.

Finally, the results presented in the paper looks promising but lack details. For instance: what is shown on the y-axis of Figure 2?

**Rebuttal**
After reading the rebuttal, I am not convinced that the new version will be more accessible.


**Time Spent Reviewing:**

2

---

> ### Author Response · Authors · 2021-08-10
> **Response to Reviewer qUuQ**
>
> We thank the reviewer qUuQ for the time and effort and appreciate him/her honestly admitting the review just an educated guess: "I must say that I have very limited knowledge on this topic so my review has to be taken with a grain of salt. "
>
> **We now answered *all* questions raised. These questions do not point to any deficiencies in our paper. Apart from a very minor suggestion (explaining an axis in a plot), these questions merely reflect the fact hat the reviewer is not familiar with this area.**
>
> **Question 1**
>
> *I must say that I have very limited knowledge on this topic so my review has to be taken with a grain of salt. Yet, I find the paper not easy to read in the many technical part. For instance, I am not sure to understand exactly what is the matrix $W(q)$ defined in section 2.4 and what are the implication of Equation (3) -> in section 5.1, it is written “as discussed in Section 2.4, a decentralized communication round can be represented by a multiplication by [...] $W(q)$“. I would have appreciated more details there.*
>
> **Response to Question 1**
>
> - Gossip matrices are widely used in the decentralized optimization literature. Almost all works on decentralized algorithms that we cite use gossip matrices or their analogue called *mixing matrices*. Such matrices have been used in the field for more than 15 years (see, for example, the seminal work "Randomized Gossip Algorithms" by Boyd, Ghosh, Prabhakar and Shah, IEEE TRANSACTIONS ON INFORMATION THEORY, VOL. 52, NO. 6, JUNE 2006), and every person working in the field is familiar with them. This is why we presented it in a compact way to save space. Our work is not an introductory book chapter or a tutorial, where defining such objects would have been fully appropriate.
>
> - Property 1 on line 100 implies that a matrix-vector multiplication with $W(q)$ can be implemented using only communication across the edges, i.e., decentralized communication. Again, this is well known.
>
> - We also give an example of $W(q)$: a Laplacian matrix of the graph. This example gives an explanation of the meaning of the parameter $\chi$. However, if a reader does not know what the Laplacian of a graph is, this will not help. Still, since our work is not an introductory text, we can't define all such well known terms. This is standard practice in any subject area.
>
> - While Equation (3) is not directly used in the lower bounds, it appears in the theoretical analysis of our algorithms. We mention it in lines 253 and 273, for example, and in the analysis in lines 562 and 583, for example.
>
> > Please note that while these questions can be asked by someone who does not work in the field, they won't be asked by people working in the field. Our work is a scientific paper written for the experts, and people with some knowledge of decentralized optimization. Our work does not aim to serve as a tutorial or an introductory book chapter on this topic. Hence, the above comments can't be seen as a deficiency of our paper.
>
> **Question 2**
>
> *What is the interest of Figure 1?*
>
> **Response to Question 2**
>
> We provided Figure 1 to make sure that the readers of our paper understand correctly how the iteration counter $k$ and communication round counter $q$ are related. We defined the concepts in the text. However, these are very important basic concepts for the understanding of many parts of the paper, and we believe a visual aid is very helpful in this regard. So, we do not want to save space by removing this figure.
>
> While it is possible to save space by removing this figure, this will lead to a slight increase in the possibility that a reader might not properly understand the relationship between $k$ and $q$. If we find more important content than this that needs to be placed into the main body of the paper, we would not hesitate to remove this figure and use this content instead. However, we think we have covered all we wanted in the main body of the paper.
>
> > Please note that, again, this question does not point to any deficiency in our paper.
>
> **Question 3**
>
> *Would not it be better to merge Theorem 2 and 3 into one?*
>
> **Response to Question 3**
>
> We are convinced that it is better to keep them separated because one is dedicated to the communication complexity lower bound, which is a key and novel bound, while the other is related to the gradient computation complexity, which coincides with the lower bound of Nesterov, which is widely known.
>
> We will add this clarification to the paper.
>
> > Please note that this question does not point to any deficiency in our paper. As we explained, this is a not a very good suggestion for saving space; and even if it was, this would be a very minor remark/suggestion at best rather than a critical comment that should have any bearing on the acceptance or not of a paper.
>
> **Question 4**
>
> *The lower bound (Theorem 1) seem to be the main contribution of the paper.*
>
> **Response to Question 4**
>
> This is *not* true. The lower bound is only about a *half* of the main contribution of our work. The second half includes the *optimal algorithms*. This is clear from our abstract (each contribution is even highlighted in italics!), introduction (we list three contributions in section 1.2, the first two of which are the key contributions, and the third, experiments, is minor), and the main text of the paper as well. We dedicate several sections to optimal algorithms. So, we do not quite understand how this point could have been missed.
>
> > This is not an issue with our paper. The reader missed a half of our contributions, which we believe is not possible to miss as we repeat it in several places.
>
> **Question 5**
>
> *The authors could comment on the impact of choosing $d=+\infty$ here? Would it change anything?*
>
> **Response to Question 5**
>
> - The limiting case $d=+\infty$ is often considered while providing the lower bounds. For example, it was used by Nesterov in his lower bounds for non-distributed optimization. It provides for a simpler and cleaner analysis. This is well known in the literature on optimal/accelerated methods, and is the subject of several books.
>
> - In contrast to Nesterov’s proof, a more rigorous proof of the same lower bound with $d < \infty$ is provided for example in the book of Nemirovski. We think, that this type of analysis could be used in the case of decentralized optimization, but the proof would become more complicated and would provide less intuition.
>
> - This question merely reveals that the reviewer is not familiar with certain standard concepts and approaches to establishing optimally of first order algorithms. Again, our work is not an introductory text, and hence it is not reasonable for us to define such concepts. Please not we rely on many advanced mathematical and optimization concepts. People without training in these areas will naturally not be able to read our paper. However, it is not possible to write a scientific paper of this type in a way that makes it accessible to a very wide audience, audience without basic knowledge in these subjects.
>
> > This is not an issue with our paper.
>
> **Question 6**
>
> *The algorithmic part is harder to understand for me. For instance, is it obvious that the multi-consensus algorithm falls into the category of algorithms used in the lower bound of Theorem 1? Also, the contribution with respect to ADOM (Kovalev et al 2021) is unclear.*
>
> **Response to Question 6**
>
> - It is written on lines 254-256 that multi-consensus algorithm only requires $T$ decentralized communication rounds per iteration, compared to the algorithm without multi-consensus. Hence, they only differ in the number of communication rounds used per iteration.
>
> - Re ADOM: We make this abundantly clear in the paper. See our introduction, Table 1, Section 1.2, and Section 5.2. If after reading these parts of the paper you still have questions, please do not hesitate to ask! You did not say what exactly is not clear to you while we believe an answer to your question is already contained in the paper. So, either you missed these parts, or your question can be clarified more so that we know what exactly is unclear to you.
>
> > This is not an issue with our paper.
>
> **Question 7**
>
> *For instance: what is shown on the $y$-axis of Figure 2?*
>
> **Response to Question 7**
>
> The $y$-axis shows the distance to the optimal solution. We did not write this explicitly, thanks for noticing this minor omission.
>
> > This is a minor suggestion.
>
> **Question 8**
>
> *To be more accessible to a non-expert like me, the assumption behind the communication model and the contribution with respect to related work could be better explained.*
>
> **Response to Question 8**
>
> - Our paper was not aimed to be accessible by non-experts. Most mathematics-heavy papers simple can't be made accessible in this way as the knowledge necessary for understanding is so vast, that it can't possibly be explained/covered within the paper. In mathematics-heavy subjects, we indeed need to stand on the shoulders of giants.
>
> - We still believe we made our paper accessible to people who work in the decentralized optimization area, and to people who work on accelerated/optimal methods and lower bounds. Going much beyond this is not possible.
>
> - Of course, every paper can be made a tad more accessible in this or that part, perhaps. But this costs real estate, and in a conference format with severe page limits, this is very difficult.
>
> - Unfortunately, it is hard for us to come up with further ideas of what should be improved in terms of exposition. Section 1.2 covers both contribution and related work in a detailed way, while Section 3.1 provides a detailed explanation of the assumptions on the decentralized algorithms, that we consider. We believe our exposition is nearly optimal given the technical depth of our contributions and the severe page limit.
>
> > This is not an issue with our paper.

---

> ### Author Response · Authors · 2021-08-21
> **Please can you respond to our rebuttal?**
>
> Dear Reviewer qUuQ,
>
> Please can you respond to our rebuttal? We have written it 12 days ago.
>
> Thanks,
>
> Authors

---

> > ### Comment · Reviewer_qUuQ · 2021-08-23
> > **Meaningless answer.**
> >
> > Dear authors,
> >
> > I apologize but your paper is not the center of the world. We all have lots of reviews to do and discussion on other papers. We started discussing the paper among reviewers but your insistence makes it more difficult.
> >
> > Regarding your answer: if I were to summarize it, the only message that I got from your answer is: "this is a paper for specialists, if you did not understand it, please read again". This will not lead me to increase my score.

---

> > > ### Author Response · Authors · 2021-08-23
> > > **Re: Meaningless answer**
> > >
> > > Dear Reviewer qUuQ,
> > >
> > > Thanks for your reply; we appreciate it.
> > >
> > > **A brief case for authors-reviewers discussion on OpenReview**
> > >
> > > We strongly believe that discussion between the reviewers and authors is the centerpiece of the scientific review process. Unlike CMT, the system used for NeurIPS conferences in the past, OpenReview *finally* allows iterative discussion rather than the linear, and we believe ineffective, *review-rebuttal-decision* approach. Discussion is absolutely crucial for obvious reasons: it allows for iterative clarification of ideas, results and criticism, which then hopefully leads to better informed decisions and recommendations by the reviewers to the Area Chair, and ultimately to a more scholarly, robust and accurate reviewing process.
> > >
> > > NeurIPS transferred from CMT to OpenReview for a few reasons, and one of the key reasons is precisely the possibility to hold reviewer-author discussions. This is why we ask for your feedback. We wish to know what you think about our rebuttal, so that we know what issues still remain to be tackled, if any. Of course, ultimately some issues may still remain even after a few rounds of communication here on OpenReview. That is fine, and to be expected. However, an iterative process will undoubtedly reduce the number of gross misunderstandings, oversights and so on.
> > >
> > > Because we believe NeurIPS reviewers are obliged to take part in such discussions, and because we have seen that for nearly two weeks we did not get any engagement, we have explicitly asked for it. We did so politely. We apologize if this seems to be inappropriate to you; our intentions were and are not to be aggressive. But we need to disagree here: we do think it is appropriate and even desirable for us to remind the reviewers to engage with us. Normally, we would expect an AC to do that.
> > >
> > > Yes, our paper is not the "center of the world". We never claimed it was. We understand the reviewers have their personal lives, professional lives, and other papers to attend to. But no engagement, in our view, equals undermining the reviewing process. Just like we believe it is the reviewers' duty to engage with the authors and their rebuttals, we believe it is our right to ask for engagement if we are getting none. Our goal is a better and more accurate reviewing process. And that takes effort. It took us a lot of effort to write our paper; we believe we have the right to defend it, to clarify what needs to be clarified, and to ask if we managed to do so or not. Thanks for your understanding. We hope this is not controversial - it should not be.
> > >
> > > You also said:
> > > >We started discussing the paper among reviewers but your insistence makes it more difficult.
> > >
> > > **We very strongly disagree with the attitude expressed in the above sentence, which might be interpreted as an attempt to undermine the review process. If this is not the case, we apologize - but it surely looks like an explicit discouragement to ask you to comment on our rebuttal.** On the other hand, we are glad that the reviewers are already discussing our work in their committee meeting with the AC.
> > >
> > > We want to engage with you and other reviewers precisely because we want to shed light on any possible misunderstandings. **We believe that at its best, a reviewing process should start with reviewers engaging with the authors in a few rounds of clarifying exchange, and only *afterwards* it should resort to a deliberation by the committee composed of reviewers and the Area Chair. Indeed, the committee needs to have the best data and input available to make a good judgement, and the reviewer-author discussion can be invaluable in that regard.
> > >
> > > **Our rebuttal**
> > >
> > > Please note that in our rebuttal we provided brief but reasonably detailed and argued replies to *all* questions raised in your original review. We would kindly request that for each question that you believe was *not* addressed properly, you provide a comment explaining why our reply did not resolve your concerns, and issues still remain. Acknowledging which issues *were* addressed properly is also important as it lets us all focus on what remains to be tackled.
> > >
> > > Your summary
> > >
> > > > "this is a paper for specialists, if you did not understand it, please read again"
> > >
> > > of our rebuttal is not correct. We did not intend to say this, and we do not believe this is how our text is best interpreted. **A big part (but not all) of our rebuttal was merely us providing a justification that the concepts you do not understand are normally not (and can't reasonable expected to be) explained in scientific papers in this field. They are basic prerequisites of the field. Just like in a paper on gradient methods one would not give the definition of a gradient, in a paper on NNs one would not normally define what a neural net is, in a paper on Bayesian inference one would not need to explain what a prior is, and in a paper on the Langevin method one would not need to explain what a stochastic differential equation really is, so in a work on decentralized optimization, certain objects are so basic and ubiquitous, that they are not defined or explained.** They form the common knowledge; and their understanding is a prerequisite for anyone reading papers in the area. We pointed out that in your case, these prerequisites were not met. This is not saying anything bad about you. We are ignorant of most fields, and would have similar lack of understanding of basic concepts in those fields. However, and this is what we tried to say, **this does not mean there is anything wrong with our paper.** **We argue that such a criticism is not justified, and should not be used to decrease the score for a paper.** This merely means that academic papers are not written for general scientific audience, which is just a matter of fact, and not something controversial. They are written for experts. You did say you are not an expert. That's all. Books, tutorials or other introductory texts are the appropriate medium for acquiring such basic knowledge. Not scientific papers.
> > >
> > > **Finally, we do not agree that our rebuttal is "meaningless". We merely defend the right of experts to assume a certain degree of familiarity with the basic concepts of their field when writing their papers. We do not wish to admit that we made a mistake because we did not. Most of the concepts you are confused about are elementary in the field. It is not a mistake to not define them.** This is the norm. And this norm exists for a good reason: one can't advance science if all papers had to develop their ideas from the axioms. Every paper needs to assume a certain type of a reader the paper is written for.
> > >
> > > **Having said that, we are happy to add some basic definitions to the supplementary material. We believe this should completely settle your concerns.**
> > >
> > >
> > > ---
> > >
> > > Finally, thanks again for your reply.
> > >
> > > Auhors

---

> > > > ### Comment · Reviewer_qUuQ · 2021-08-24
> > > > **I agree to discuss but you seem to refuse to discuss.**
> > > >
> > > > As you said clearly in your answer, openreview is a place for discussion. Yet, to me discussing does not just mean taking but also means listening. I have read the reviews of all, your answers to all and my impression is that you disagree with everything that was suggested by the reviewers (essentially your comments start with "We are fully convinced that the reviewer failed to recognize [...]" or end with *this is not an issue of our paper". You have the right to disagree.
> > > >
> > > > Concerning my review: as said before, I will not fight for acceptance or reject of your paper because I am not the specialist here. Yet, I do believe that a good paper should make itself accessible to a reasonable audience. I do not think that the examples that you cite in your answers are relevant: 99% of the NeurIPS audience probably knows what is a gradient, a NN, or a prior. SDE might be more technical but is also a basic concept. Thus, I do not think that your paper is accessible to a very large audience. This might not be issue but this is a fact. And to make a paper more accessible, it is not a new section in the appendix that is needed but also proper references (to this section or to other books).
> > > >
> > > > As an outsider, I mostly gave some advice on what I did not understand in your paper so that you can improve it. If you refuse to see them, I am sorry.

---

### Official Review · Reviewer_isKt · 2021-07-14

**Rating:** 5
**Confidence:** 5

**Summary:**

This work studies the problem of smooth and strongly convex decentralized optimization problems in time-varying networks and it is relevant to the conference. The paper provides both lower bound and optimal algorithms for decentralized optimization that combines several existing works, then apply them in time-varying networks. The paper prove its algorithm is optimal, in sense of the computation and communication complexity. There are some light experiments associated with the paper, thus the practical performance of the approach is somehow validated.

**Limitations And Societal Impact:**

Have the authors adequately addressed the limitations and potential negative societal impact of their work?
Yes

**Main Review:**

The main contribution of this paper is to offer both lower bound and rate optimal algorithms for smooth and strongly convex decentralized optimization problems in time-varying networks. The problem is well-motivated and the writing is clear. There are some light simulation results thus the proposed algorithm seems work from both the theory and simulations perspective.

Decentralized distributed optimization in the strongly convex and smooth regime is relatively well studied. In particular, [1] established lower decentralized communication and local computation complexities for solving this problem, and proposed an optimal algorithm called MSDA in the case when an access to the dual oracle is assumed. Under a primal oracle, current state of the art includes the near-optimal algorithms [2], and a recently proposed optimal algorithm OPAPC [3]. The main contribution of this work is an extension of above works into time-varying networks. Thus the obtained lower bound is not surprise. Build up on that, the authors further introduce a rate optimal algorithm which match the rate of lower bound and the results are validated by light simulations. However, the simulation is based on logistic regression which is way too simple in current deep learning era. I don't see any points of an experiments using only 100 data points per node.

In short, the paper is technically sound and the developments are clear. The derived algorithm and rates seems to be a useful contribution to the literature on decentralized composite convex optimization, showing a modest improvement over the state of the art. ​However, the paper is still a combine-then-twist work based on top of several existing works and could be strengthened by demonstrating more significant results instead of incremental. It could be accepted if there is room, but should not warrant a strong accept due to the lack of novelty and experiments.

[1] Scaman, K., Bach, F., Bubeck, S., Lee, Y. T., and Massoulié, L. (2017). Optimal algorithms for smooth and strongly convex distributed optimization in networks. In international conference on machine learning, pages 3027–3036. PMLR.

[2] Dvinskikh, D. and Gasnikov, A. (2019). Decentralized and parallelized primal and dual accelerated methods for stochastic convex programming problems. arXiv preprint arXiv:1904.09015.

[3] Kovalev, D., Salim, A., and Richtárik, P. (2020). Optimal and practical algorithms for smooth and strongly convex decentralized optimization. Advances in Neural Information Processing Systems, 33.

---
after thoroughly and carefully read the paper and response after rebuttal, the best I can do is to keep my score unchanged (if not lower it). Basically, the response said nothing, and the paper is a trivial combination work with only limited experimental results.

**Time Spent Reviewing:**

10

---

> ### Author Response · Authors · 2021-08-10
> **General response to Reviewer isKt**
>
> We thank reviewer isKt for the time and effort. We appreciated that the reviewer described our work positively by saying:
> - "The main contribution of this paper is to offer both lower bound and rate optimal algorithms for smooth and strongly convex decentralized optimization problems in time-varying networks."
> - "The problem is well-motivated and the writing is clear."
> - "There are some light simulation results thus the proposed algorithm seems work from both the theory and simulations perspective."
> - "In short, the paper is technically sound and the developments are clear. "
> - "The derived algorithm and rates seems to be a useful contribution to the literature on decentralized composite convex optimization, showing a modest improvement over the state of the art."
>
> As far as we understand, there are 2 main concerns raised by the reviewer:
>
> 1) significance of the theoretical results, and
> 2) simplicity of the experiments.
>
> Below we address these concerns and believe that our work does not suffer from the issues mentioned. We hope the reviewer accepts the explanations. If that is the case, no major issues remain, and we hence kindly request that the reviewer re-evaluates our work.
>
> We believe the contributions in our work are in the 8-10 range in terms of score.

---

> ### Author Response · Authors · 2021-08-10
> **Response to Issue 1: Significance of theoretical results**
>
> **Issue 1: Significance of theoretical results**
>
> *The main contribution of this work is an extension of above works into time-varying networks. Thus the obtained lower bound is not surprise. Build up on that, the authors further introduce a rate optimal algorithm which match the rate of lower bound and the results are validated by light simulations.*
>
> *... However, the paper is still a combine-then-twist work based on top of several existing works and could be strengthened by demonstrating more significant results instead of incremental.*
>
> **Response to Issue 1**
>
> We are fully convinced that the reviewer failed to recognize the main theoretical developments of the paper. The claims that
>
> - "...thus the obtained lower bound is not surprise"
> - "the paper is still a combine-then-twist work based on top of several existing works"
> - "could be strengthened by demonstrating more significant results instead of incremental"
>
> are unsupported by *any* evidence. We have some questions:
>
> - Why exactly is the lower bound not a surprise, and why should it be a surprise? How does it follow that a work that is an "extension" of some previous works is automatically "not a surprise"?
> - Even if we granted that our bounds are not "surprising" (we do not; and we do not believe eliciting surprise is what research is about), this does not mean that it was easy to obtain. These are two very different things. For example, often in research there are conjectures that everyone believes to be true, yet no one knows how to prove them. If a proof is found, a reviewer will certainly not belittle the work by saying that the result was not a surprise - indeed, it was not, since everyone knew it. What was not know was the proof...
> - Why is our work a "combine-then-twist" work? What does this even mean? We have never heard this phrase before. Why is this bad? Again, can you give evidence of your claims? Your claim is not intelligible to us; we do not know what it means. No explanation was given. How can we then defend our work? Criticism of this type should not be accepted at venues such as NeurIPS.
> - Why are our results not significant? What is incremental about them? In some sense, *all* research is incremental. We all build on prior works in one way or another. The word here is used as a negative adjective. Why exactly? Are you suggesting our results were easy to obtain? Again, this is generic criticism which we do not understand as no explanation was given. How can we make our results "more significant"? No suggestions were given. What is more significant than obtaining an optimal algorithm, and first lower bounds for a long studied problem? Surely, we can't obtain a method that is "more" optimal... Please can you offer any concrete suggestions on what results in the area we study would you deem to be less incremental and more significant?
>
> Let us address the above points by providing some explanations.
>
> Our algorithm ADOM+ is by no means just an "extension" of MSDA or OPAPC. We note that the reviewer simply just claimed this, without providing any evidence. (We believe that claims made without evidence can be rejected without evidence.)
> Please note that
> - Our method uses a different reformulation of the original problem, and the realization that this formulation would yield results is a contribution on its own, clearly showing that our results can't possible be just a simple extension of the previous works mentioned.
> - Our method uses a different mechanism for achieving linear convergence under decentralized communication over time-varying networks. Again, this is a major difference.
> - There are many implementation details we describe which result in a complex algorithm, and a complex theoretical analysis significantly different from the analyses of previous works. This was acknowledged by reviewer 6mM6. We suspect that reviewer isKt did not delve deep into the details of our theoretical results and how they were obtained due (the reviewer also self-reported that he/she did not spend enough time on our paper).
>
> Our algorithm ADOM is the *first accelerated (optimal) algorithm which uses a dual based oracle*.
> - There were attempts to provide dual based oracles, such as "Panda: A dual linearly converging method for distributed optimization over time-varying undirected graphs" or "Optimal distributed convex optimization on slowly time-varying graphs". All these attempts failed due to the lack of provable acceleration, or due to unrealistic assumptions.
> - The main issue with applying Nesterov acceleration to the dual reformulation of the problem is that it has to work under a variable metric, or a variable function. We found an elegant solution to this open question. Moreover, we were able to adapt it to a saddle point reformulation (ADOM+), which is already challenging on its own. None of this are simple extensions of previous works.
>
> Our lower bound development was not trivial either, far from it! This development alone, we believe, is a major contribution! Indeed, we give the first lower bound in the time-varying network regime. If this as easy to do, it would have been done long ago. Note that:
> - First we had to find lower bounds for a problem of minimization of an arbitrary sequence of smooth and strongly convex functions with the same minimum. Only after we did this, we got an understanding of how to construct a corresponding sequence of networks and then combine this all with the lower bounds of Nesterov.
> - It is also worth mentioning that the lower bounds of Scaman et al. appeared in 2017, while the problem of decentralized optimization over time-varying networks has been studied even longer. This suggests that it was difficult to obtain lower bounds in the fixed network regime. However, neither lower bounds nor optimal algorithms were developed until now for the problem of time varying networks! We close the gap in the literature which as open for a long time. This is a clear indication that our results can't be trivial. In fact, we believe our work is resolves a major open problem in the field.
>
> With all that said, our theoretical results by no means incremental.
> - While our results are strong, correct and novel, it is possible that the reviewer might have gotten tricked by our well-written and clear explanation of the main theoretical ideas. We indeed spent a lot of time in explaining our ideas as intuitively as possible. However, this does not mean, and is not the case, that the results were easy to obtain! The story, as we tell it, is not the story describing the actual research we had to undertake, and the many pitfalls and blind alleys we had to avoid and learn from to achieve our results. Instead, the story we portray is an attempt to make the results sound as intuitively understandable as possible with the knowledge we have *after* we have obtained them. Good quality of writing and explanation in our paper should not be mistaken for the results being of an incremental nature.
> - The reviewer did not specify what exactly is incremental in our work and why. Again, claims made without evidence can be dismissed without evidence. This more true in research than in ordinary life.

---

> ### Author Response · Authors · 2021-08-10
> **Response to Issue 2: Experiments**
>
> **Issue 2: Experiments**
>
> *There are some light simulation results thus the proposed algorithm seems work from both the theory and simulations perspective.*
>
> *Build up on that, the authors further introduce a rate optimal algorithm which match the rate of lower bound and the results are validated by light simulations. However, the simulation is based on logistic regression which is way too simple in current deep learning era. I don't see any points of an experiments using only 100 data points per node.*
>
> **Response to Issue 2**
>
> Experiments are *not* the main contribution of our paper.
>
> - Our theoretical work stands on its own without any experiments at all, just like many great experimental-only works can be accepted (and celebrated) despite not being supported by any theory. In other words, one should *not* have a higher bar for theoretical works in this regard, and require that they be accompanied by (any, let alone strong) experiments, just like we do not require a practical work to be supported by theory to be accepted. So, we are convinced that the experiments we provide should simply be seen as an additional bonus, shedding additional light on certain properties of our method in relation to the provided theory
>
> - State-of-the-art works like [1], [3] and "ADOM: Accelerated decentralized optimization method for time-varying networks" also provide experiments with 100 samples per node, while [2] does not provide experiments at all. Given the fact, that our paper has a solid theoretical contribution (and this is *the key* contribution), and closes an important gap between lower bounds and algorithms in the area of decentralized optimization over time-varying networks, it should be absolutely valid that experimental section is not the strongest part of our paper.
>
> - Suggesting that our experiments are "too simple in current deep learning era" is ill-conceived. each experiment has a goal, and the experiment is designed to test that goal. It is simply not true that all experiments should always be *large* or *complicated* to be useful. Quite the opposite is true. If simple experiments can test a particular goal, then clearly they should be preferred to unnecessarily complicated experiments. What is too simple about our experiments and why more complicated experiments are needed? No justification is given. Saying that we we live deep learning era is nor a justification. Our work is not about deep learning, and we do not even consider nonconvex regime. So, this suggestion is simply out of place.

---

> ### Author Response · Authors · 2021-08-21
> **Please can you respond to our rebuttal?**
>
> Dear Reviewer isKt,
>
> Please can you respond to our rebuttal? We have written it 12 days ago.
>
> Thanks,
>
> Authors

---

### Official Review · Reviewer_6mM6 · 2021-07-15

**Rating:** 8
**Confidence:** 4

**Summary:**

This paper studies the decentralized optimization ove time-varying networks. Lower bounds for smooth and strongly convex problems are proved at the first time and optimal algorithms that attain these lower bounds are provided.

**Limitations And Societal Impact:**

This paper discusses the limitations in Section 5.3. I am interested in whether the primal-dual framework can be applied to nonstrongly convex problems. This may be another limination.

**Main Review:**

Originality: This paper establishes the first lower bounds for decentralized optimization over time-varying networks. The task is more challenging than the case of fixed network. I think it is a novel contribution. Optimal algorithms are provided which attain these lower bounds. The presented method ADOM+ is a novel extention of the one in Kovalev 2020.

Quality: The theory is technically solid. Both lower bounds and upper bounds are provided. This is a complete work. The authors discuss both the strengths and weaknesses of the proposed methods.

Clarity: This paper is written well.

Significance: Decentralized optimization over time-varying networks is an important task in signal processing, automatic control, and federated learning. This paper provides the state-of-the-art results to this task. I think the contributions are important.

Comments:

1. The third assumption on line 102 means that the elements of each column of W(q) sum to 0, which is equivalent to the assumption that I-W(q) is column stochastic. What about the rows? We often assume the double stochastic mixing matrices for undirected graphs, and column (or row) stochastic ones for directed graphs. I suggest the authors to check it carefully whether the row stochastic assumption is really avoided.

2. The presented ADOM+ algorithm is not implementable because step 5 dependes on $y^{k+1}$, which is computed in step 8, whle step 8 dependes on $x^{k+1}$, which is computed in step 5. This is a little problem and can be easily addressed. However, I suggest to give a detailed explanation after the presentation of the algorithm.

3. In the experiment, the authors only test the performance of ADOM+ with multi-consensus. According to my experience, the multi-consensus are not working in practice. I suggest to compare ADOM+ with multi-consensus, ADOM+ without multi-consensus, Acc-GT with multi-consensus, and Acc-GT without multi-consensus in the final version. This is not a compulsion in the rebuttal because the time is limited to redo the experiment.

4. Both the ADOM+ algorithm and its analysis are complex. I have read the proof, but I cannot get the ideas. I suggest to provide more intuitions and explanations of the proofs.

**Time Spent Reviewing:**

6 hours

---

> ### Author Response · Authors · 2021-08-10
> **Response to Reviewer 6mM6**
>
> We greatly appreciate that reviewer 6mM6 made a significant effort to understand the details of our paper and gave valuable and very meaningful comments, which we address here.
>
> **Comment 1**
>
> *The third assumption on line 102 means that the elements of each column of $W(q)$ sum to $0$, which is equivalent to the assumption that $I-W(q)$ is column stochastic. What about the rows? We often assume the double stochastic mixing matrices for undirected graphs, and column (or row) stochastic ones for directed graphs. I suggest the authors to check it carefully whether the row stochastic assumption is really avoided.*
>
> **Response to Comment 1**
>
> As you correctly noted, $I-W(q)$ is indeed a column stochastic matrix. However, the second assumption on line 101 means that elements of each row of $W(q)$ sum to $0$ as well, and hence $I-W(q)$ is row stochastic as well. In other words, we assume double stochasticity of the mixing matrix. We will clarify this in the next version of the paper.
>
> **Comment 2**
>
> *The presented ADOM+ algorithm is not implementable because step 5 depends on $y^{k+1}$, which is computed in step 8, while step 8 depends on $x^{k+1}$, which is computed in step 5. This is a little problem and can be easily addressed. However, I suggest to give a detailed explanation after the presentation of the algorithm.*
>
> **Response to Comment 2**
>
> You are right, this problem can be easily addressed, for example, by finding an equation for $y^{k+1}$ from line 8, plugging it into line 5 and finding an explicit equation for $x^{k+1}$. We did not do it due to lack of space. We will add a paragraph with a detailed explanation in the updated version of our paper.
>
> **Comment 3**
>
> *In the experiment, the authors only test the performance of ADOM+ with multi-consensus. According to my experience, the multi-consensus are not working in practice. I suggest to compare ADOM+ with multi-consensus, ADOM+ without multi-consensus, Acc-GT with multi-consensus, and Acc-GT without multi-consensus in the final version. This is not a compulsion in the rebuttal because the time is limited to redo the experiment.*
>
> **Response to Comment 3**
>
> This is a valuable comment. In the current version of the paper we indeed performed only a comparison of algorithms supported by state-of-the-art theory. We will provide more experiments in the final version of the paper, including more practical versions of the algorithms without multi-consensus.
>
> **Comment 4**
>
> *Both the ADOM+ algorithm and its analysis are complex. I have read the proof, but I cannot get the ideas. I suggest to provide more intuitions and explanations of the proofs.*
>
> **Response to Comment 4**
>
> We were able to include the basic intuition about the algorithm development, but a more detailed description was not performed due to lack of space. The proof and the algorithm are indeed complex. However, the idea of the proof comes form the ideas and mechanisms used in the algorithm development, which we explained in detail. It is of course possible to write a more detailed paragraph which covers both ideas used in the algorithm development and corresponding proof techniques that come from these ideas. If our paper is accepted, we will use an extra page to provide such detailed paragraph.
>
> **Comment 5**
>
> *This paper discusses the limitations in Section 5.3. I am interested in whether the primal-dual framework can be applied to nonstrongly convex problems. This may be another limitation.*
>
> **Response to Comment 5**
>
> Thank you for pointing this out. This is indeed an interesting question. One possible way to deal with it is to add a small enough regularization $||x - x^0||^2$ to the original problem, which depends on the desired precision. We will also think about other alternative ways of solving this issue, and will add a corresponding comments in the final version of the paper.

---

> > ### Comment · Reviewer_6mM6 · 2021-08-20
> > **Response to the rebuttal**
> >
> > Thanks for the rebuttal. I keep my score unchanged.

---

> > > ### Author Response · Authors · 2021-08-21
> > > **Thanks and please champion our paper**
> > >
> > > Dear Reviewer 6mM6,
> > >
> > > Thanks for responding to our rebuttal. **We would like to kindly ask you to champion our paper in the reviewer/committee discussion phase.** Please defend our work.
> > >
> > > You will see that we addressed issues mentioned by other reviewers as well. We would be happy to defend our work ourselves, but the other reviewers are not responding to our rebuttal here on OpenReview. So, we do not know whether they are convinced by our arguments or not. If they told us what remains to be explained, we would happily do so.
> > >
> > > Best regards,
> > >
> > > Authors

---

### Official Review · Reviewer_jE4H · 2021-07-17

**Rating:** 5
**Confidence:** 4

**Summary:**

This paper considers the problem of minimizing the finite-sum of smooth and strongly convex functions in decentralized time-varying networks. The authors establish the first lower bounds on the number of decentralized communication rounds and the number of local computations required to find an approximate solution. They also designed two optimal algorithms that attain these lower bounds. Experiment results of the proposed methods are also presented.

**Limitations And Societal Impact:**

There are no foreseeable negative societal impact of this work.

**Main Review:**

Originality: The obtained result of lower bound is new, though its proof appears to be a straightforward extension of that in Scaman et al. (2017). The multi-concensus procedure is new to the best of my knowledge.

Quality: The paper is technically sound. All the claims are equipped with solid proof. The technical results are correct.

Clarity: The paper is well written and organized.

Significance: I have some concerns on the significance and potential impact of this paper. First, the authors claim that the obtained lower bound is for time-varying networks, but I cannot see how the time-varying property plays a role in the main theorems. Also, what is the lower bound for time-invariant networks? In general, we are expecting the lower bound for time-varying networks to be larger than the lower bound for time-invariant networks, and we are interested in how the time-varying property affects the bound. Second, it is interesting to see that multi-consensus leads to optimal local computation rounds. However, it might be only of theoretical interest because it significantly increases the communication rounds (although the bound on communication rounds remains the same). Since in practice the communication cost is usually the bottleneck, the practical impact of multi-consensus strategy is rather limited.

**Time Spent Reviewing:**

4

---

> ### Author Response · Authors · 2021-08-10
> **Response to Reviewer jE4H**
>
> We thank the reviewer jE4H for the time and effort.
> We now address some parts of the review.
>
> **Issue 1: "proof appears to be a straightforward extension of that in Scaman et al."**
>
> Please can you elaborate why you believe the extension is straightforward? You did not offer any reasoning, and only offered this claim. We dispute this claim; the work is not a straightforward extension of Scaman et al - this is far from being the case.
>
> - Although, it is subjective whether a proof is deemed to be straightforward or not, we think that the novelty of our proof compared to the proof of Scaman et al. is more or less the same as the novelty of the proof of Scaman et al. compared to the proof of Nesterov. In our view, the work of Scaman was a non-trivial extension of previous work that did not consider decentralized setup, and our work is a non-trivial extension from fixed to time-varying networks.
> -  The problem of decentralized optimization over time-varying networks has been studied for a number of years, but neither lower bounds nor optimal algorithms have been proposed until now. Moreover, during the work on this paper we first proposed a lower complexity bound for a sequence of smooth and strongly convex functions with the same minimum, which is already very different from the proof of Nesterov. Then we understood how to implement a sequence of time-varying networks with similar structure.
> These facts suggest that our proof is not that straightforward as one could think.
>
> > This is actually not an issue.
>
> **Issue 2: "I cannot see how the time-varying property plays a role in the main theorems."**
>
> - Theorems 1, 2 and 3 make this very clear as they explicitly talk about a sequence of graphs (the sequence corresponds to the network evolving in time, as we explain in the paper). So, we dispute that the time-varying property is hard to see here. The opposite is true: it is obvious.
> -  Theorems 4 and 6 talk about our method ADOM+ and ADOM+ with multi-consensus, respectively, which clearly includes time-varying gossip matrices $W(q)$. Moreover, the theorem involves the time-varying network condition number $\chi$ defined in Section 2.4. Again, the time-varying nature of the result is clear.
> - Theorem 5 talks about ADOM with multi-consensus. We explain the time-varying nature of the setup for this method in Section 5.2. The theorem involves the time-varying network condition number $\chi$ defined in Section 2.4.
>
> Because of the above, we do not really see how it is possible for the reviewer to miss the role of the time-varying nature of the graphs in the theorems.
>
> - Moreover, we gave an overview of our proof techniques on lines 160-167. In particular we explain, how we construct the sequence of time-varying networks, and how it is used in the proof.
> - We suggest the reviewer jE4H to read the proof in the appendix for more details.
>
> > This is actually not an issue.
>
> **Issue 3: "Also, what is the lower bound for time-invariant networks?"**
>
> - We mention lower bounds for time-invariant networks proposed by Scaman et al. on lines 59-62. After reading Scaman et al., one can conclude that when the network is fixed in time and symmetric, lower communication complexity bound is $O(\sqrt{\chi\kappa}\log 1/\epsilon)$ compared to $O(\chi\sqrt{\kappa}\log 1/\epsilon)$ in our paper. Hence, our lower bound is indeed larger by the factor $\sqrt{\chi}$.
> - However, our paper does not cover the case of time-invariant networks and hence we do not include such comparison. This should by no means influence the rating of our paper.
>
> > This is actually not an issue.
>
> **Issue 4: "it is interesting to see that multi-consensus leads to optimal local computation rounds. However, it might be only of theoretical interest because it significantly increases the communication rounds (although the bound on communication rounds remains the same)."**
>
> Theory from our paper shows that multi-consensus does *not* increase the number of communication rounds (so, the reviewer is wrong here!) but reduces the number of local computation rounds. Hence, there is no surprise that it leads to both optimal communication and local computation complexity.
>
> > This is actually not an issue.
>
> **Summary**
>
> The main concern of reviewer jE4H is significance of our work.
>
> By reading the comments of the reviewer and our replies to these comments, one can conclude that what the reviewer believes are "issues" with our work are not issues with our work, but merely issues with the reviewer not performing a careful reading of our paper. This is quite rare in our experience.
>
> > None of the issues are issues!
>
> Hence, we suggest that reviewer jE4H significantly increases his/her score.

---

> ### Author Response · Authors · 2021-08-21
> **Please can you respond to our rebuttal?**
>
> Dear Reviewer jE4H,
>
> Please can you respond to our rebuttal? We have written it 12 days ago.
>
> Thanks,
>
> Authors

---

### Author Response · Authors · 2021-08-10
**Message to all Reviewers and the AC**

Dear reviewers and the AC,

Thanks for taking time to read our work and for providing comments. A lot of positive was said about our paper:

**By Reviewer jE4H (score 5):**

> "The paper is technically sound."

> "All the claims are equipped with solid proof."

> "The technical results are correct."

> "The paper is well written and organized."

> "The obtained result of lower bound is new"

> "The multi-concensus procedure is new to the best of my knowledge."

> "...it is interesting to see that multi-consensus leads to optimal local computation rounds."

**By Reviewer 6mM6 (score 8):**

> "This paper establishes the first lower bounds for decentralized optimization over time-varying networks. The task is more challenging than the case of fixed network. I think it is a novel contribution."

> "Optimal algorithms are provided which attain these lower bounds. The presented method ADOM+ is a novel extention of the one in Kovalev 2020."

> "The theory is technically solid. Both lower bounds and upper bounds are provided. This is a complete work."

> "The authors discuss both the strengths and weaknesses of the proposed methods."

> "This paper is written well."

> "Decentralized optimization over time-varying networks is an important task in signal processing, automatic control, and federated learning. This paper provides the state-of-the-art results to this task. I think the contributions are important."

**By Reviewer isKt (score 5):**

> "The main contribution of this paper is to offer both lower bound and rate optimal algorithms for smooth and strongly convex decentralized optimization problems in time-varying networks. "

> "The problem is well-motivated and the writing is clear."

> "There are some light simulation results thus the proposed algorithm seems work from both the theory and simulations perspective."

> "The main contribution of this work is an extension of above works into time-varying networks. "

> "Build up on that, the authors further introduce a rate optimal algorithm which match the rate of lower bound and the results are validated by light simulations."

> "In short, the paper is technically sound and the developments are clear. "

> "The derived algorithm and rates seems to be a useful contribution to the literature on decentralized composite convex optimization, showing a modest improvement over the state of the art."

---

We received 4 reviews. Unfortunately, only one reviewer (Reviewer 6mM6) offered useful and expert commentary on our paper. This reviewer gave our work the score of 8. Although reviewers isKt and jE4H offered many positive comments, for which we are very thankful, the criticism they raised was either generic, unsubstantiated, or simply not valid - and we hope this is apparent from our respective rebuttals. So, we believe either these reviewers should increase their scores substantially in response to our rebuttal, or otherwise that the AC should ignore these reviews in his/her judgment. The last reviewer, Reviewer qUuQ, offered an "educated guess" only. No valid criticism was raised there either, as we explain in our rebuttal. However, we are thankful that despite this reviewer not comprehending our work, he/she viewed our work positively.

**We kindly request Reviewer 6mM6 and the AC to carefully read our rebuttals to all reviews and defend our work in the reviewer discussion phase. THANK YOU!**

Kind regards,

Authors


​

---

> ### Author Response · Authors · 2021-08-19
> **Please can you respond to our rebuttals?**
>
> Dear reviewers,
>
> Please can we kindly ask you to respond to our rebuttals?
>
> Thanks in advance!
>
> authors of Paper9754

---

### Author Response · Authors · 2021-08-15
**Please can you let us know if you've read our rebuttal and whether we addressed your concerns?**

Dear reviewers,

We did not receive any feedback on our author response yet from any reviewer.

Please can you let us know whether you've read our rebuttal and whether we addressed your concerns?

If we did not, please let us know what we failed to address appropriately.

Thanks!!

Authors of Paper 9754

---

### Decision · Program_Chairs · 2021-09-27

**Decision:**

Accept (Poster)

**Comment:**

The paper presents a significant contribution to distributed optimization over time-varying networks. In particular it establishes a lower-bound on complexity  by identifying a specific pattern of time-varying networks, and proposes two schemes (one, ADOM+, requiring access to dual gradients, and another one avoiding the need to access such dual gradients) that are proven to match this lower bound. This thus gives a rather complete treatment of the problem of distributed convex optimization for time-varying networks as those considered in the paper.
While the paper's analysis is made for specific assumptions (eg on the type of time varying networks) and the lower bound is of a worst-case nature, the authors clarified these points, and proposed to develop discussions of these and other points in the final version which should put the paper in a correct perspective.